# TFG-Flow: Training-free Guidance in Multimodal Generative Flow

**Haowei Lin**[1][*], **Shanda Li**[2][*], **Haotian Ye**[3]
**Yiming Yang**[2], **Stefano Ermon**[3], **Yitao Liang**[1], **and Jianzhu Ma**[4]
[1]Peking University, [2]Carnegie Mellon University, [3]Stanford University, [4]Tsinghua University

## Abstract

Given an unconditional generative model and a predictor for a target property (e.g., a classifier), the goal of training-free guidance is to generate samples with desirable target properties without additional training. As a highly efficient technique for steering generative models toward flexible outcomes, training-free guidance has gained increasing attention in diffusion models. However, existing methods only handle data in continuous spaces, while many scientific applications involve both continuous and discrete data (referred to as multimodality). Another emerging trend is the growing use of the simple and general flow matching framework in building generative foundation models, where guided generation remains under-explored. To address this, we introduce **TFG-Flow**, a novel training-free guidance method for multimodal generative flow. TFG-Flow addresses the curse-of-dimensionality while maintaining the property of unbiased sampling in guiding discrete variables. We validate TFG-Flow on four molecular design tasks and show that TFG-Flow has great potential in drug design by generating molecules with desired properties.[1]

## 1 Introduction

Recent advancements in generative foundation models have demonstrated their increasing power across a wide range of domains (Reid et al., 2024; Achiam et al., 2023; Abramson et al., 2024). In particular, diffusion-based foundation models, such as Stable Diffusion (Esser et al., 2024) and SORA (Brooks et al., 2024) have achieved significant success, catalyzing a new wave of applications in areas such as art and science. As these models become more prevalent, a critical question arises: how can we steer these foundation models to achieve specific properties during inference time?

One promising direction is using classifier-based guidance (Dhariwal & Nichol, 2021) or classifier-free guidance (Ho & Salimans, 2022), which typically necessitate training a specialized model for each conditioning signal (e.g., a noise-conditional classifier or a text-conditional denoiser). This resource-intensive and time-consuming process greatly limits their applicability. Recently, there has been growing interest in *training-free guidance for diffusion models*, which allows users to steer the generation process using an off-the-shelf differentiable target predictor without requiring additional model training (Ye et al., 2024). A target predictor can be any classifier, loss, or energy function used to score the quality of the generated samples. Training-free guidance offers a flexible and efficient means of customizing generation, holding the potential to transform the field of generative AI.

Despite significant advances in generative models, most existing training-free guidance techniques are tailored to diffusion models that operate on continuous data, such as images. However, extending generative models to jointly address both discrete and continuous data—referred to as multimodal data (Campbell et al., 2024)—remains a critical challenge for broader applications in scientific fields (Wang et al., 2023). One key reason this expansion is essential is that many real-world problems involve multimodal data, such as molecular design, where both discrete elements (e.g., atom types) and continuous attributes (e.g., 3D coordinates) must be modeled together. To address this, recent generative foundation models have increasingly adopted the flow matching framework (Esser et al.,

---

[*]These two authors contributed equally to the paper. Correspondence to Haowei Lin (`linhaowei@pku.edu.cn`) and Jianzhu Ma (`majianzhu@tsinghua.edu.cn`).

[1]Code is available at `https://github.com/linhaowei1/TFG-Flow`.

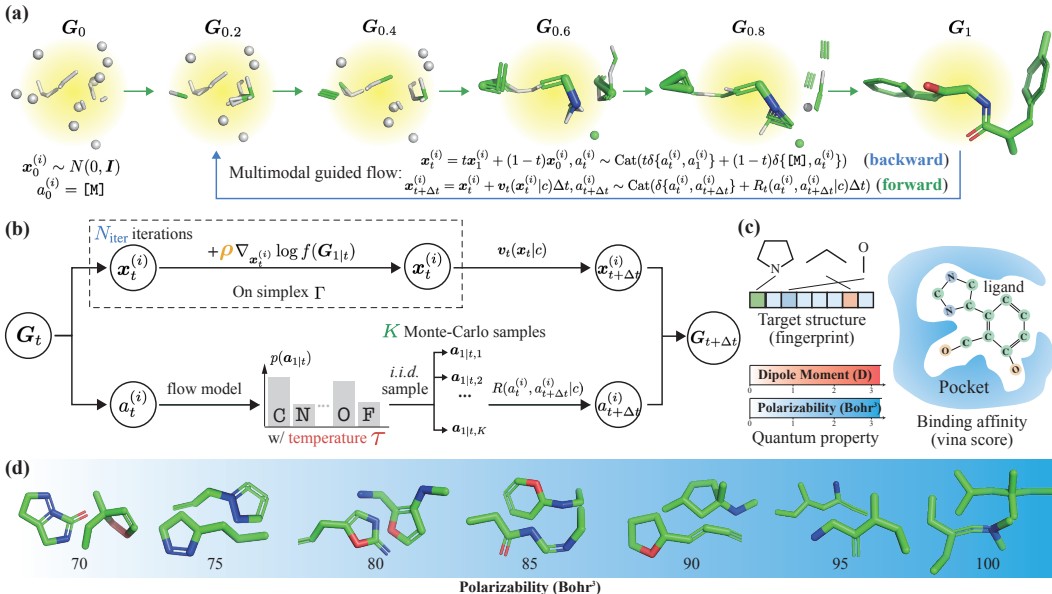

Figure 1: **(a) Multimodal guided flow.** The backward flow (blue arrow) is constructed with linear interpolation between observed data $G_1$ and sampled noise $G_0$; the forward flow (green arrows) is simulated by conditional velocity $v_t(x_t^{(i)}|c)$ and conditional rate matrix $R_t(a_t^{(i)}, a_{t+\Delta t}^{(i)}|c)$. **(b) The illustration of TFG-Flow.** TFG-Flow guides the forward flow on each step $G_t$, which consists of a gradient-based guidance for continuous part $X_t$ and an importance sampling based guidance for discrete part $a_t$; **(c) Some examples of guidance targets. (d) Samples guided by TFG-Flow targeted at polarizability.** The targeted value is at bottom of the samples.

2024), prized for its simplicity and general applicability to both data types. Multiflow, recently introduced by Campbell et al. (2024) on protein co-design problem, presents a promising foundation for tackling multimodal generation via Continuous Time Markov Chains (Anderson, 2012).

Unfortunately, guided generation within the flow matching framework remains relatively underexplored, due to the inherent differences between guiding continuous and discrete data. This paper investigates the problem of training-free guidance in multimodal flow matching, with a specific focus on its application to inverse molecular design (Zunger, 2018). Inverse molecular design is a challenging task that involves generating molecules that meet specific target properties, such as a desired level of polarizability. Our method, TFG-Flow, construct a guided flow that aligns with target predictor while preserving the marginals of unguided flow, effectively enabling plug-and-play guidance (Theorems 3.1 and 3.2). Unlike continuous variables, where gradient information is inherently informative, discrete variables cannot be adjusted continuously. Naive guidance methods that estimate transition probabilities between discrete states suffer from the *curse of dimensionality*, a long-lasting problem in machine learning that is computationally intractable. To address this, TFG-Flow devises a consistent Monte-Carlo sampling approach that reduces the complexity from exponential to logarithmic for discrete guidance (Theorem 3.4), while leveraging a partial derivative-based guidance mechanism for continuous variables that preserves geometric invariance (Theorem 3.5).

We apply TFG-Flow to various inverse molecular design tasks. When targeted to quantum properties, TFG-Flow is able to generate more accurate molecules than existing training-free guidance methods for continuous diffusion (with an average relative improvement of +20.3% over the best baseline). When targeted to specific molecular structures, TFG-Flow improves the similarity to target structures of unconditional generation by more than 20%. When targeted at multiple properties, TFG-Flow outperforms conditional multimodal flow significantly. We also apply TFG-Flow to pocket-based drug design tasks, where TFG-Flow can guide the flow to generate molecules with more realistic 3D structures and better binding energies towards the protein binding sites compared to the baselines.

Our main contributions are summarized as follows:

- We present TFG-Flow, a novel approach for guiding multimodal flow models toward target properties in a training-free manner (via off-the-shelf time-independent property functions).

- TFG-Flow sets a solid theoretical foundation for multimodal guided flow, addresses the curse of dimensionality, and maintains geometric invariance via theorems 3.1∼3.5.
- Experiments reveal that TFG-Flow is effective and efficient for designing 3D molecules with desired properties, opening up new opportunities to explore the chemical space.

## 2 PRELIMINARY

In this section, we provide readers with the essential knowledge required for 3D molecule generation using a multimodal flow model.

### 2.1 SO(3) INVARIANT 3D MOLECULE MODELING AND GENERATION

**3D molecular representations.** Let $\mathcal{A}$ denote the set of all atom types plus a "mask" state $[\texttt{M}]$ which indicates an undetermined atom type and will be useful in our generative model. A molecule with $n$ atoms can be represented as $\boldsymbol{G} = (\boldsymbol{X}, \boldsymbol{a}) \in \mathcal{G}$, where $\mathcal{G} = \mathbb{R}^{3 \times n} \times \mathcal{A}^n$, $\boldsymbol{X}$ denotes the atomic coordinates, and $\boldsymbol{a}$ represents the atom types. We also use $\boldsymbol{x}^{(i)}$ and $a^{(i)}$ to denote the coordinates and type of the $i$-th atom, respectively.

**Invariant probabilistic modeling for 3D molecules.** Invariance is a crucial inductive bias in modeling 3D geometry of molecules (Gilmer et al., 2017). Molecular systems exist in three-dimensional Euclidean space, where the group associated with rotations is known as $\mathrm{SO}(3)$. In this work, we are interested in the distribution $p(\boldsymbol{X}, \boldsymbol{a})$ whose marginal of $\boldsymbol{X}$ satisfies the following property:

$$\text{SO(3)-\textbf{invariance}:} \quad p(\boldsymbol{X}) = p(\boldsymbol{S}\boldsymbol{X}) \quad \text{for any } \boldsymbol{S} \in \mathrm{SO}(3). \tag{1}$$

Here, $\mathrm{SO}(3)$ denotes the set of all the rotation matrices in 3D space. Intuitively, the definition requires the likelihood of a molecular structure to be invariant with respect to any transformation in $\mathrm{SO}(3)$.

Additionally, molecular representations need to be invariant to translations. To ensure this, we always project the atomic coordinates to $\Gamma = \{\boldsymbol{X} \mid \boldsymbol{x}^{(1)} + \cdots + \boldsymbol{x}^{(n)} = \boldsymbol{0}\}$ via mean subtraction: $\mathrm{Project}_\Gamma(\boldsymbol{X}) := \boldsymbol{X} - \frac{1}{n}\sum_{i=1}^{n} \boldsymbol{x}^{(i)} \mathbf{1}_n^\top$, where $\mathbf{1}_n$ denotes the $n$-dimensional all-one row vector.

**Equivariant graph neural networks.** In this work, we follow existing research and employ equivariant neural networks to model the $\mathrm{SO}(3)$-invariant distribution of molecule structures (Hoogeboom et al., 2022; Xu et al., 2022). Specifically, we apply Equivariant Graph Neural Networks (EGNNs) to process molecular representations (Satorras et al., 2021). EGNN takes molecular representations $\boldsymbol{G} = (\boldsymbol{X}, \boldsymbol{a})$ as its input and translate the atom type $\boldsymbol{a}^{(i)}$ into $d$-dimensional embedding $\boldsymbol{h}^{(i)} \in \mathbb{R}^d$ for each atom. In each layer, EGNN updates atomic coordinates and embeddings as follows:

$$\boldsymbol{m}^{(i,j)} \leftarrow \Phi_\mathrm{m}(\boldsymbol{h}^{(i)}, \boldsymbol{h}^{(j)}, \|\boldsymbol{x}^{(i)} - \boldsymbol{x}^{(j)}\|^2; \boldsymbol{\theta}_\mathrm{m}); \tag{2}$$

$$\boldsymbol{h}^{(i)} \leftarrow \Phi_\mathrm{h}\left(\boldsymbol{h}^{(i)}, \sum_{j=1}^{n} \boldsymbol{m}^{(i,j)}; \boldsymbol{\theta}_\mathrm{h}\right); \tag{3}$$

$$\boldsymbol{x}^{(i)} \leftarrow \boldsymbol{x}^{(i)} + \sum_{j=1}^{n} \left(\boldsymbol{x}^{(i)} - \boldsymbol{x}^{(j)}\right) \Phi_\mathrm{x}(\boldsymbol{m}^{(i,j)}; \boldsymbol{\theta}_\mathrm{x}), \tag{4}$$

In the above, $\boldsymbol{m}^{(i,j)}$ denotes intermediate message from atom $i$ to $j$, and $\Phi_\mathrm{m}, \Phi_\mathrm{h}, \Phi_\mathrm{x}$ are learnable modules parameterized by $\boldsymbol{\theta}_\mathrm{m}, \boldsymbol{\theta}_\mathrm{h}, \boldsymbol{\theta}_\mathrm{x}$, respectively. We will formally prove that this architecture, along with the design of our algorithm, leads to $\mathrm{SO}(3)$-invariant distributions.

### 2.2 MULTIMODAL FLOW MODEL

Multiflow is a multimodal flow model originally developed for protein co-design (Campbell et al., 2024). In this work, we adapt it for small molecule design. Multiflow constructs a probability flow $p_t(\boldsymbol{G}_t)$ for $t \in [0,1]$, where $p_0(\boldsymbol{G}_0) = p_\mathrm{noise}(\boldsymbol{G}_0)$ and $p_1(\boldsymbol{G}_1) = p_\mathrm{data}(\boldsymbol{G}_1)$. During inference, one first samples $\boldsymbol{G}_0 \sim p_0$ and then generates a sequence of $\boldsymbol{G}_t$ values by simulating the flow.

**Conditional flow, velocity, and rate matrix.** At the core of the sampling process is the multimodal conditional flow[2] $p_{t|1}(\boldsymbol{G}_t|\boldsymbol{G}_1)$. By design, this flow can be factorized over both the number of atoms and their modalities:

$$p_{t|1}(\boldsymbol{G}_t|\boldsymbol{G}_1) := \prod_{i=1}^{n} p_{t|1}(\boldsymbol{x}_t^{(i)}|\boldsymbol{x}_1^{(i)})p_{t|1}(a_t^{(i)}|a_1^{(i)}). \tag{5}$$

In the above, the flows $p_{t|1}(\boldsymbol{x}_t^{(i)}|\boldsymbol{x}_1^{(i)})$ (continuous) and $p_{t|1}(a_t^{(i)}|a_1^{(i)})$ (discrete) are defined such that they transport the noise distribution to the data distribution following straight-line paths, which is known as *rectified flow* (Liu et al., 2022): $\boldsymbol{x}_t^{(i)}|\boldsymbol{x}_1^{(i)} \sim \mathcal{N}(t\boldsymbol{x}_1^{(i)}, (1-t)^2\boldsymbol{I})$ and $a_t^{(i)}|a_1^{(i)} \sim \mathrm{Cat}(t\delta\{a_t^{(i)}, a_1^{(i)}\} + (1-t)\delta\{[\texttt{M}], a_t^{(i)}\})$. Here, $\delta\{b, b'\}$ is the Kronecker delta, which is 1 when $b = b'$ and 0 otherwise, and $\mathrm{Cat}$ denotes a Categorical distribution over $\mathcal{A}$.

With the definition, we can build in closed form the conditional velocity $\boldsymbol{v}_{t|1}(\cdot|\boldsymbol{x}_1^{(i)})$ and conditional rate matrix $R_{t|1}(\cdot, \cdot|a_1^{(i)})$ which characterize the dynamics of the above conditional distributions:

$$\boldsymbol{v}_{t|1}(\boldsymbol{x}_t^{(i)}|\boldsymbol{x}_1^{(i)}) = \frac{\boldsymbol{x}_1^{(i)} - \boldsymbol{x}_t^{(i)}}{1-t}; \quad R_{t|1}(a_t^{(i)}, b|a_1^{(i)}) = \frac{\delta\{b, a_1^{(i)}\}\delta\{a_t^{(i)}, [\texttt{M}]\}}{1-t}. \tag{6}$$

**Deriving the unconditional flow.** With Eq. (7), the conditional velocity and rate matrix can then be used to derive their unconditional counterparts by sampling from $p_{1|t}(\boldsymbol{G}_1|\boldsymbol{G}_t)$, which the flow model learns during training. In Multiflow and our implementation, the flow model takes $\boldsymbol{G}_t$ as the input and predict the expectation of $\boldsymbol{x}_{1|t}$ for the continuous part as well as the distribution $p_{1|t}(\boldsymbol{a}_1|\boldsymbol{G}_t)$ for the discrete part. Note that directly modeling the expectation $\mathbb{E}_{1|t}[\boldsymbol{x}_{1|t}|\boldsymbol{G}_t]$ instead of the distribution $p_{1|t}(\boldsymbol{x}_{1|t}|\boldsymbol{G}_t)$ is not a problem since $\boldsymbol{v}_{t|1}$ is linear $w.r.t.$ $\boldsymbol{x}_1$, which will be discussed in App. C.4.

$$\boldsymbol{v}_t(\boldsymbol{x}_t^{(i)}) = \mathbb{E}_{p_{1|t}(\boldsymbol{G}_1|\boldsymbol{G}_t)}\left[\boldsymbol{v}_{t|1}(\boldsymbol{x}_t^{(i)}|\boldsymbol{x}_1^{(i)})\right]; \quad R_t(a_t^{(i)}, b) = \mathbb{E}_{p_{1|t}(\boldsymbol{G}_1|\boldsymbol{G}_t)}\left[R_{t|1}(a_t^{(i)}, b|a_1^{(i)})\right]. \tag{7}$$

The unconditional continuous and discrete flows satisfy the Fokker-Planck and Kolmogorov Equations: $\partial_t p_t = -\nabla \cdot (\boldsymbol{v}_t p_t)$ and $\partial_t \boldsymbol{p}_t = \boldsymbol{p}_t R_t$.[3] Therefore, during inference, starting from an initial sample $\boldsymbol{G}_0 \sim p_0$[4], one can iteratively estimate the unconditional velocity and rate matrix at the current time step $t$ by sampling from the learned distribution $p_{1|t}(\boldsymbol{G}_1|\boldsymbol{G}_t)$. The Fokker-Planck and Kolmogorov Equations can then be simulated to generate $\boldsymbol{G}_{t+\Delta t}$ for the next time step, ultimately resulting in $\boldsymbol{G}_1 \sim p_1$ which approximates the data distribution.

## 3 METHODOLOGY

**Problem setup.** Our objective is to develop an effective, *training-free* method to guide an unconditional multimodal flow model to generate samples with desired properties. Formally, let $c$ denote our target property, and assume we have a time-independent target predictor $f_c(\boldsymbol{G}_1) = p(c|\boldsymbol{G}_1)$ that quantifies how well a given molecule $\boldsymbol{G}_1$ satisfies the target property $c$. We also have access to a model $g(\boldsymbol{G}_t)$, which represents $p_{1|t}(\boldsymbol{G}_1|\boldsymbol{G}_t)$ of a flow model $\{p_t(\boldsymbol{G}_t)\}_{t\in[0,1]}$, i.e., it takes $\boldsymbol{G}_t$ as input and returns the sample/likelihood of $\boldsymbol{G}_1$, as defined in Sec. 2.

In this section, we present our construction of a multimodal guided flow $\{p_t(\boldsymbol{G}_t|c)\}_{t\in[0,1]}$ that satisfies $p_1(\boldsymbol{G}_1|c) = p_{\mathrm{data}}(\boldsymbol{G}_1|c)$. Then we derive an algorithm which simulates the constructed flow using the target predictor $f_c(\boldsymbol{G}_1)$, the flow model $g(\boldsymbol{G}_t)$, and a suitable velocity and rate matrix.

### 3.1 CONSTRUCTING THE GUIDED FLOW

Recall that the critical factor in the inference with flow models is to derive the velocity and rate matrix in the Fokker-Planck and Kolmogorov Equations so that we can simulate the sample over time. For guided generation, we need to address the following two challenges:

---

[2]*Conditional flow* refers to the flow $p_{t|1}(\boldsymbol{G}_t^{(i)}|\boldsymbol{G}_1^{(i)})$ conditioned on the clean data $\boldsymbol{G}_1^{(i)}$. This contrasts with the desired *unconditional* flow for sampling, *i.e.*, $p_t(\boldsymbol{G}_t^{(i)})$, which can be derived using Eq. (7) from the conditional flow.

[3]In the Kolmogorov Equation, the discrete flow should be seen as a row vector $\boldsymbol{p}_t = (p_t(a))_{a\in\mathcal{A}}$.

[4]In practice, we sample $\boldsymbol{x}_0^{(i)} \sim \mathcal{N}(\boldsymbol{0}, \boldsymbol{I}_3)$ and $a_0^{(i)} = [\texttt{M}]$ independently for each $i$. As discussed in Sec. 2.1, we additionally project $\boldsymbol{X}_0$ to the simplex $\Gamma$.

- Can we construct the guided flow $p_t(\boldsymbol{G}_t|c)_{t\in[0,1]}$ and derive the corresponding guided velocity and rate matrix in the Fokker-Planck and Kolmogorov Equations of it?

- Can we efficiently estimate the guided velocity and rate matrix and simulate the Fokker-Planck and Kolmogorov Equations?

**Constructing appropriate guided flows.** While infinitely many flows $p_t(\boldsymbol{G}_t|c)_{t\in[0,1]}$ satisfy $p_1(\boldsymbol{G}_1|c) = p_{\text{data}}(\boldsymbol{G}_1|c)$, we seek to specify a joint distribution of $(\{\boldsymbol{G}_t\}_{t\in[0,1]}, c)$ which can be simulated via minimal adaptation of the original flow when incorporating the guidance $f_c(\boldsymbol{G}_1)$. An ideal guided flow should maintain three key characteristics: (1) preserve the overall structure and dynamics of the original flow model by maintaining the flow marginals $\{\boldsymbol{G}_t\}_{t\in[0,1]}$; (2) align with the conditional distribution $f_c(\boldsymbol{G}_1) = p(c|\boldsymbol{G}_1)$; and (3) ensure independence between $\{\boldsymbol{G}_t\}_{t\in[0,1)}$ and the condition $c$ when the clean data $\boldsymbol{G}_1$ is observed. These desiderata allows us to minimally modify the original flow while effectively steering the generation process towards the desired molecular properties. We formulate these intuitive criteria mathematically and prove the existence of a guided flow satisfying these conditions in the following theorem:

**Theorem 3.1** (Existence of the guided flow (*informal*))**.** *Let $\mathcal{G}$ be the space of molecular representations and $\mathcal{C}$ be a finite set which includes all the values of our target property. Given a $\mathcal{G}$-valued process $\{\boldsymbol{G}_t\}_{t\in[0,1]}$ modeled by flow $\{p_t^{\mathrm{G}}(\boldsymbol{G}_t)\}_{t\in[0,1]}$ and a function $f_c(\boldsymbol{G}_1)$ which defines a valid distribution over $\mathcal{C}$ for any $\boldsymbol{G}_1 \in \mathcal{G}$, there exists a joint distribution $p$ of random variables $(\{\boldsymbol{G}_t\}_{t\in[0,1]}, c)$ that satisfies the following:*

- ***Preservation of flow marginals:*** *For any $t \in [0,1]$, $p_t(\boldsymbol{G}_t) = p_t^{\mathrm{G}}(\boldsymbol{G}_t)$.*

- ***Alignment with target predictor:*** *For any $c \in \mathcal{C}$ and $\boldsymbol{G}_1 \in \mathcal{G}$, $p(c|\boldsymbol{G}_1) = f_c(\boldsymbol{G}_1)$.*

- ***Conditional independence of trajectory and target:*** *For any $t \in [0,1)$, $\boldsymbol{G}_t$ and $c$ are conditionally independent given $\boldsymbol{G}_1$, i.e., $p_{t|1}(\boldsymbol{G}_t|\boldsymbol{G}_1, c) = p_{t|1}(\boldsymbol{G}_t|\boldsymbol{G}_1)$.*

The formal version of Theorem 3.1 based on measure theory, along with its proof, can be found in App. B.1. We note that the conditional independence of trajectory and target is particularly appealing because it ensures that the central tool in flow model – the conditional flow – remains unchanged under guidance. Specifically, the conditional independence of trajectory and target ensures that for a Multiflow model (defined in Sec. 2.2) under guidance, the guided conditional flow $p_{t|1}(\boldsymbol{G}_t|\boldsymbol{G}_1, c)$ still satisfies the factorization property as defined in Eq. (5):

$$p_{t|1}(\boldsymbol{G}_t|\boldsymbol{G}_1, c) := \prod_{i=1}^{n} p_{t|1}(\boldsymbol{x}_t^{(i)}|\boldsymbol{x}_1^{(i)}) p_{t|1}(a_t^{(i)}|a_1^{(i)}). \tag{8}$$

Furthermore, in the guided flow, the flow marginals are preserved, hence the conditional velocity and rate matrix for the continuous and discrete conditional flows are the same as those defined in Eq. (6).

**Finding the guided velocity and rate matrix.** Parallel to Sec. 2.2, it now suffices to derive the unconditional guided velocity and rate matrix for the continuous and discrete flows based on the conditional counterpart. We present our result in the following theorem:

**Theorem 3.2** (Guided velocity and rate matrix (*informal*))**.** *Consider a continuous flow $p_t(\boldsymbol{x}_t)$ with conditional velocity $\boldsymbol{v}_{t|1}(\boldsymbol{x}_t|\boldsymbol{x}_1)$ and a discrete flow model $p_t(a_t)$ with conditional rate matrix $R_{t|1}(a_t, b|a_1)$. Then the guided flows $p_t(\boldsymbol{x}_t|c)$ and $p_t(a_t|c)$, defined via the construction in Theorem 3.1, can be generated by $\boldsymbol{v}_t(\boldsymbol{x}_t|c)$ and $R_t(a_t, j|c)$ via Fokker-Planck Equation $\partial_t p_t = -\nabla \cdot (\boldsymbol{v}_t p_t)$ and Kolmogorov Equation $\partial_t \boldsymbol{p}_t = \boldsymbol{p}_t R_t$, where*

$$\textit{Guided velocity:} \qquad \boldsymbol{v}_t(\boldsymbol{x}_t|c) = \mathbb{E}_{p_{1|t}(\boldsymbol{x}_1|\boldsymbol{x}_t, c)} \left[ \boldsymbol{v}_{t|1}(\boldsymbol{x}_t|\boldsymbol{x}_1) \right], \tag{9}$$

$$\textit{Guided rate matrix:} \quad R_t(a_t, b|c) = \mathbb{E}_{p_{1|t}(a_1|a_t, c)} \left[ R_{t|1}(a_t, b|a_1) \right]. \tag{10}$$

The formal statement and proof are in App. B.2. The conditional velocity $\boldsymbol{v}_{t|1}$ and conditional rate matrix $R_{t|1}$ for Multiflow are defined in Eq. (6). Thus to simulate the guided flow $p_t(\boldsymbol{G}_t|c)$, it suffices to sample from $p_{1|t}(\cdot|\boldsymbol{G}_t, c)$ so that we can estimate the velocity and rate matrix. In the subsequent subsections, we present our sampling methods for the discrete and continuous parts of the flow.

## 3.2 DISCRETE GUIDANCE: ADDRESSING CURSE OF DIMENSIONALITY

Recall that the learned Multiflow model can output $p_{1|t}(\boldsymbol{a}_1|\boldsymbol{G}_t)$ for $\boldsymbol{a}_1 \in \mathcal{A}^n$. Furthermore, one can show via Theorem 3.1 that $p_{1|t}(\boldsymbol{a}_1|\boldsymbol{G}_t, c) \propto p_{1|t}(\boldsymbol{a}_1|\boldsymbol{G}_t)f_c(\boldsymbol{G}_1)$ where $f_c(\cdot)$ is the target predictor. Thus, $R_t(a_t^{(i)}, j|c)$ can be exactly calculated based on Eq. (10). However, we note that this is computationally intractable because of the curse of dimensionality: The complexity of exactly calculating the expectation is $\mathcal{O}(|\mathcal{A}|^n)$, growing exponentially with the number of atoms $n$.

Therefore, it is natural to resort to Monte-Carlo approaches to estimate $R_t(a_t^{(i)}, b|c)$. One straightforward idea is to consider importance sampling. We show in Proposition B.4 that the guided rate matrix defined in Theorem 3.2 satisfies

$$R_t(a_t^{(i)}, b|c) = \mathbb{E}_{p_{1|t}(\boldsymbol{G}_1|\boldsymbol{G}_t)} \left[ \frac{p(c|\boldsymbol{G}_1)}{p(c|\boldsymbol{G}_t)} R_{t|1}(a_t^{(i)}, b|a_1^{(i)}) \right]. \tag{11}$$

However, this approach requires access to $p(c|\boldsymbol{G}_t)$, i.e., a time-dependent target predictor. This requires additional classifier training and prohibit the use of off-the-shelf predictors. Thus, importance sampling is not directly applicable and we build our method based on the following observation:

**Proposition 3.3.** *For $t \in [0, 1)$, the guided rate matrix defined in Theorem 3.2 satisfies*

$$R_t(a_t^{(i)}, b|c) = \frac{\mathbb{E}_{p_{1|t}(\boldsymbol{G}_1|\boldsymbol{G}_t)} \left[ f_c(\boldsymbol{G}_1) R_{t|1}(a_t^{(i)}, b|a_1^{(i)}) \right]}{\mathbb{E}_{p_{1|t}(\boldsymbol{G}_1|\boldsymbol{G}_t)} \left[ f_c(\boldsymbol{G}_1) \right]}. \tag{12}$$

Proposition 3.3 indicates that we can generate $K$ samples from $p_{1|t}(\boldsymbol{G}_1|\boldsymbol{G}_t)$ and obtain a good estimation of the guided rate matrix based on the known conditional rate matrix $R_{t|1}(a_t^{(i)}, b|a_1^{(i)})$ and target predictor $f_c(\boldsymbol{G}_1)$. We conduct theoretical analysis on the approximation error of this method in Theorem 3.4. We show that $K = \mathcal{O}\left(\log(n|\mathcal{A}|)\right)$ samples suffice to accurately approximate $R_t(a_t^{(i)}, b|c)$ for any $i \in \{1, \cdots, n\}$ and $b \in \mathcal{A}$ with high probability. This is in sharp contrast to the $\mathcal{O}(|\mathcal{A}|^n)$ complexity of exact expectation calculation, avoiding the curse-of-dimensionality issue. Furthermore, our method does not require access to unknown quantities such as $p(c|\boldsymbol{G}_t)$ in Eq. (11). We show in the experiments that this approach can effectively simulate the discrete component of the multimodal guided flow.

**Theorem 3.4.** *Let $\boldsymbol{G}_{1|t,1}, \cdots, \boldsymbol{G}_{1|t,K} \sim$ i.i.d. $p_{1|t}(\cdot|\boldsymbol{G}_t)$. Define the estimation of $R_t(a_t^{(i)}, b|c)$ as*

$$\hat{R}_t(a_t^{(i)}, b|c) = \sum_{k=1}^{K} f_c(\boldsymbol{G}_{1|t,k}) R_{t|1}(a_t^{(i)}, b \mid a_{1|t,k}^{(i)}) \bigg/ \sum_{k=1}^{K} f_c(\boldsymbol{G}_{1|t,k}). \tag{13}$$

*Assume $\underline{f_c} = \inf_{\boldsymbol{G} \in \mathcal{G}} f_c(\boldsymbol{G}) > 0$. Given any $\varepsilon \in (0, \underline{f_c}/2)$, $\delta \in (0, 1)$, if $K = \Theta\left(\frac{1}{\varepsilon^2} \log \frac{n|\mathcal{A}|}{\delta}\right)$, then*

$$\mathbb{P}\left( \sup_{i \in \{1, \cdots, n\}, \, b \in \mathcal{A}} \left| \hat{R}_t(a_t^{(i)}, b|c) - R_t(a_t^{(i)}, b|c) \right| < \varepsilon \right) \geq 1 - \delta \tag{14}$$

## 3.3 CONTINUOUS GUIDANCE: BUILDING EQUIVARIANT GUIDED ODE

For the continuous component, recall that the learned Multiflow model predicts the expectation $\mathbb{E}_{1|t}[\boldsymbol{X}_1|\boldsymbol{G}_t]$ given $\boldsymbol{G}_t = (\boldsymbol{X}_t, \boldsymbol{a}_t)$ as the input, while we need $\mathbb{E}_{1|t}[\boldsymbol{X}_1|\boldsymbol{G}_t, c]$ to estimate the velocity for the guided flow as demonstrated by Theorem 3.2. To bridge the gap, our method aims to generate $\hat{\boldsymbol{G}}_t = (\hat{\boldsymbol{X}}_t, \boldsymbol{a}_t)$ such that $p_{1|t}(\cdot \mid \hat{\boldsymbol{G}}_t) \approx p_{1|t}(\cdot \mid \boldsymbol{G}_t, c)$, so that we can generate $\boldsymbol{X}_{1|t}$ from the given flow model and estimate the velocity.

Intuitively, $\hat{\boldsymbol{X}}_t$ needs to contain more information on the target $c$ to impose the guidance on $\boldsymbol{X}_{1|t}$. Motivated by training-free guidance methods developed for continuous diffusion models (Bansal et al., 2023; He et al., 2024; Ye et al., 2024), we employ a gradient ascent strategy to align the modeled expectation of $\boldsymbol{X}_{1|t}$ with the target $c$. The gradient is then propagated to $\boldsymbol{X}_t$, enabling us to derive the desired $\hat{\boldsymbol{X}}_t$. Specifically, we simulate the following iterative process:

$$\boldsymbol{X}_t \leftarrow \boldsymbol{X}_t + \rho_t \nabla_{\boldsymbol{X}_t} \log f(\mathbb{E}[\boldsymbol{X}_1|\boldsymbol{X}_t, \boldsymbol{a}_t]), \tag{15}$$

where $\mathbb{E}[\boldsymbol{X}_1|\boldsymbol{X}_t, \boldsymbol{a}_t]$ is approximated by the flow model, and $\rho_t$ is a hyperparameter to control the strength of guidance. We run the update for $N_{\text{iter}}$ steps to obtain $\hat{\boldsymbol{G}}_t = (\hat{\boldsymbol{X}}_t, \boldsymbol{a}_t)$. Then we can estimate the guided velocity based on Theorem 3.2 and simulate the Fokker-Planck Equation of the guided flow.

Additionally, we note that the continuous variables in our application represents coordinates in the 3D space. Translations and rotations in the 3D space should not change the likelihood of the variable. To ensure this property, we adopt two techniques: (a) As discussed in Sec. 2.1, we ensure that coordinates after translations are mapped to the same molecular representations by projecting the coordinates to the simplex $\Gamma = \{\boldsymbol{X} \mid \boldsymbol{x}^{(1)} + \cdots + \boldsymbol{x}^{(n)} = \boldsymbol{0}\}$. When generating $\hat{\boldsymbol{X}}_t$ via Eq. (15) in our continuous guidance algorithm, we implement the following update:

$$\boldsymbol{X}_t \leftarrow \text{Project}_\Gamma \left( \boldsymbol{X}_t + \rho_t \nabla_{\boldsymbol{X}_t} \log f(\mathbb{E}[\boldsymbol{X}_1|\boldsymbol{X}_t, \boldsymbol{a}_t]) \right). \tag{16}$$

(b) For rotation transformations, the flow $\{p_t(\boldsymbol{X}_t, \boldsymbol{a}_t|c)\}$ needs to be $SO(3)$-invariant *w.r.t.* $\boldsymbol{X}_t$, as defined in Sec. 2.1. We use Equivariant Graph Neural Networks (EGNN) as the backbone for the model $g_\theta$ so that any $SO(3)$ transformation on $\boldsymbol{X}_t$ will lead to the same transformation on $\boldsymbol{X}_{1|t} \sim p_{1|t}(\cdot|\boldsymbol{G}_t)$, i.e., the model ensures equivariant sampling (Satorras et al., 2021). We mathematically show that our techniques guarantees $SO(3)$-invariance of the simulated flow:

**Theorem 3.5** ($SO(3)$-invariance (*proof in App. B.1*))**.** *Assume the target predictor $f_c(\boldsymbol{G})$ is $SO(3)$-invariant, the model $g_\theta(\boldsymbol{G})$ is $SO(3)$-equivariant, and the distribution of $\boldsymbol{X}_0$ is $SO(3)$-invariant, then for any $t \in [0, 1]$, $p_t(\boldsymbol{X}_t|c)$ in the simulated multimodal guided flow is $SO(3)$-invariant.*

### 3.4 TFG-FLOW: PUTTING THEM ALL TOGETHER

We combine the techniques of guiding the discrete component from Sec. 3.2 and the continuous component from Sec. 3.3 into our proposed TFG-Flow, which is illustrated in Figure 1(b) and presented as pseudo code in Algorithm 1. Specifically, TFG-Flow guides the molecules $\boldsymbol{G}_t$ at each time step. For the discrete component $\boldsymbol{a}_t$, it samples $K$ Monte Carlo samples to estimate the guided rate matrix $R(a_t^{(i)}, a_{t+\Delta t}^{(i)}|c)$ using Eq. (13). For the continuous component $\boldsymbol{x}_t$, it simulates the ODE in the simplex $\Gamma$ for $N_{\text{iter}}$ iterations with Eq. (15) and obtain the guided velocity $\boldsymbol{v}_t(\boldsymbol{x}_t^{(i)}|c)$.

From an implementation standpoint, we also introduce a temperature coefficient $\tau$ to adjust the strength of discrete guidance by controlling the uncertainty of the target predictor $f_c(\boldsymbol{G}) = \text{softmax}(f(\boldsymbol{G}; \psi)/\tau)$, where $f(\cdot; \psi)$ is typically an EGNN parametrized by $\psi$ trained to classify or regress the property $c$ of a molecule $\boldsymbol{G}$. In the experiments, we set $K = 512$ and $N_{\text{iter}} = 4$ to balance computational efficiency, and tune $\rho, \tau$ via grid search (see App. D for details).[5]

## 4 EXPERIMENTS

In this paper, we explore the application of TFG-Flow across four types of guidance targets: single quantum property, combined quantum properties, structural similarity, and target-aware drug design quality. Quantum properties are examined using QM9 dataset (Ramakrishnan et al., 2014), while structural similarity is assessed on both QM9 and the larger GEOM-Drug dataset (Axelrod & Gomez-Bombarelli, 2022). The target-aware drug design quality is tested using CrossDocked2020 dataset (Francoeur et al., 2020). The baseline implementation and datasets details are in App. D.

### 4.1 QUANTUM PROPERTY GUIDANCE

**Dataset and models.** We follow the inverse molecular design literature (Bao et al., 2022; Hoogeboom et al., 2022) to establish this benchmark. The QM9 dataset is split into training, validation, and test sets, comprising 100K, 18K, and 13K samples, respectively. The training set is further divided into two non-overlapping halves. To prevent reward hacking, we use the first half to train a property prediction network for guidance and an unconditional flow model, while the second half is used to train another property prediction network that serves as the ground truth oracle, providing labels for MAE computation. All three networks share the same architecture as defined by EDM, with an EGNN as the backbone. In inference, we use 100 Euler sampling steps for the flow model.

---

[5]For simplicity, we did not schedule $\rho$ and $\tau$ (e.g., by increasing or decreasing them over time).

Table 1: How generated molecules align with the target quantum property on QM9 dataset. The results for Upper bound, #Atoms, Lower bound, Cond-EDM, EEGSDE are copied from Bao et al. (2022), and the results for training-free methods for diffusion are copied from Ye et al. (2024). The results of Cond-Flow and TFG-Flow are averaged over 3 seeds, with detailed results reported in App. E. Among all training-based or training-free methods, the best MAE for the same target is in **bold**, while the second best MAE is underlined.

| Baseline category | Method | Model category | $C_v$ | $\mu$ | $\alpha$ | $\Delta\varepsilon$ | $\varepsilon_{\text{HOMO}}$ | $\varepsilon_{\text{LUMO}}$ |
|---|---|---|---|---|---|---|---|---|
| Reference | Upper bound | \ | 6.87 | 1.61 | 8.98 | 1464 | 645 | 1457 |
| | #Atoms | | 1.97 | 1.05 | 3.86 | 886 | 426 | 813 |
| | Lower bound | | 0.040 | 0.043 | 0.09 | 65 | 39 | 36 |
| Training-based | Cond-EDM | Continuous Diffusion | 1.065 | 1.123 | 2.78 | 671 | 371 | 601 |
| | EEGSDE | | **0.941** | **0.777** | **2.50** | **487** | **302** | **447** |
| | Cond-Flow | Multimodal Flow | 1.52 | 0.962 | 3.10 | 805 | 435 | 693 |
| Training-free | DPS | | 5.26 | 63.2 | 51169 | 1380 | 744 | NA |
| | LGD | | 3.77 | 1.51 | 7.15 | 1190 | 664 | 1200 |
| | FreeDoM | Continuous Diffusion | 2.84 | 1.35 | 5.92 | 1170 | 623 | 1160 |
| | MPGD | | 2.86 | 1.51 | 4.26 | 1070 | 554 | 1060 |
| | UGD | | 3.02 | 1.56 | 5.45 | 1150 | 582 | 1270 |
| | TFG | | 2.77 | 1.33 | 3.90 | 893 | 568 | 984 |
| | TFG-Flow | Multimodal Flow | **1.75** | **0.817** | **2.32** | **804** | **364** | **941** |

Table 2: How generated molecules align with the target quantum property. Our training-free guidance TFG-Flow significantly outperforms the conditional flow (Cond-Flow) which requires condition-labelled data for training.

| Method | MAE1↓ | MAE2↓ | Method | MAE1↓ | MAE2↓ | Method | MAE1↓ | MAE2↓ |
|---|---|---|---|---|---|---|---|---|
| $C_v$ ($\frac{\text{cal}}{\text{mol}}$K), $\mu$ (D) | | | $\Delta\varepsilon$ (meV), $\mu$ (D) | | | $\alpha$ (Bohr$^3$), $\mu$ (D) | | |
| Cond-Flow | $4.96^{\pm0.07}$ | $1.57^{\pm0.01}$ | Cond-Flow | $53.5^{\pm0.71}$ | $1.57^{\pm0.00}$ | Cond-Flow | $9.33^{\pm0.01}$ | $1.54^{\pm0.01}$ |
| TFG-Flow | $\mathbf{2.36}^{\pm0.01}$ | $\mathbf{1.13}^{\pm0.04}$ | TFG-Flow | $\mathbf{46.4}^{\pm0.016}$ | $\mathbf{0.853}^{\pm0.05}$ | TFG-Flow | $\mathbf{4.62}^{\pm0.01}$ | $\mathbf{1.24}^{\pm0.05}$ |

**Guidance target.** We study guided generation of molecules on 6 quantum mechanics properties, including polarizability $\alpha$ (Bohr$^3$), dipole moment $\mu$ (D), heat capacity $C_v$ ($\frac{\text{cal}}{\text{mol}}$K), highest orbital energy $\epsilon_{\text{HOMO}}$ (meV), lowest orbital energy $\epsilon_{\text{LUMO}}$ (meV) and their gap $\Delta\epsilon$ (meV). Denote the property prediction network as $\mathcal{E}$, then our guidance target is given by an energy function $f(\boldsymbol{G}) := \exp(-\|\mathcal{E}(\boldsymbol{G}) - c\|_2^2)$, where $c$ is the target property value. For combined properties, we combine the energy functions linearly with equal weights following Bao et al. (2022) and Ye et al. (2024).

**Evaluation metrics.** We use the Mean Absolute Error (MAE) to measure guidance performance. We generate 4,096 molecules for each property to perform the evaluation following Ye et al. (2024). For completeness, we also report more metrics such as validity, novelty and stability in App. D.2.

**Baselines.** As the training-free guidance for multimodal flow is not studied before, the most direct baselines are training-free guidance methods in continuous diffusion. We compare TFG-Flow with DPS (Chung & Ye, 2022), LGD (Song et al., 2023), FreeDoM (Yu et al., 2023), MPGD (He et al., 2024), UGD (Bansal et al., 2023), and TFG (Ye et al., 2024) which treat both atom types and coordinates as continuous variables and perform guidance using gradient information. We also compare TFG-Flow with training-based baselines, such as the conditional diffusion model (Cond-EDM (Hoogeboom et al., 2022)) and conditional multimodal flow (Cond-Flow), and EEGSDE (Bao et al., 2022), which applies both training-based guidance and conditional training. We also provide referential baselines following Hoogeboom et al. (2022) (see App. D.3 for details).

**Results analysis.** The experimental results for single and multiple quantum properties are shown in Tables 1 and 2, respectively. First of all, our TFG-Flow exhibits the best guidance performance among all training-free guidance methods, with an average relative improvement of +20.3% over the best training-free guidance method TFG in continuous diffusion. Also, we note that TFG-Flow is comparable with Cond-Flow on single property guidance and outperforms Cond-Flow significantly on multiple property guidance, while Cond-Flow requires conditional training. These results justify that the discrete guidance is more effective for the discrete variables in nature. It is also noteworthy that all the training-free guidance methods underperform EEGSDE by a large margin, suggesting a large room for improvement in training-free guidance.

Table 3: How generated molecules align with the target structure. The Tanimoto similarity results are averaged over 3 random seeds.

| Dataset | Baseline | Similarity↑ |
|---|---|---|
| QM9 | Upper bound | $0.164^{\pm0.004}$ |
| | TFG-Flow | $0.290^{\pm0.007}$ |
| | *Rel. improvement* | *+76.83%* |
| GEOM-Drug | Upper bound | $0.170^{\pm0.001}$ |
| | TFG-Flow | $0.208^{\pm0.002}$ |
| | *Rel. improvement* | *+22.35%* |

Table 4: Evaluation of generated molecules for 100 protein pockets of CrossDocked2020 test set. The results of AR, Pocket2Mol, TargetDiff are from Guan et al. (2023).

| | Vina score↓ | QED↑ | SA↑ |
|---|---|---|---|
| Test set | $-6.87^{\pm2.30}$ | $0.48^{\pm0.20}$ | $0.73^{\pm0.14}$ |
| AR | $-6.20^{\pm1.25}$ | $0.50^{\pm0.11}$ | $0.67^{\pm0.09}$ |
| Pocket2Mol | $-7.23^{\pm2.04}$ | $\mathbf{0.57}^{\pm0.09}$ | $\mathbf{0.75}^{\pm0.07}$ |
| TargetDiff | $-7.32^{\pm2.08}$ | $0.48^{\pm0.15}$ | $0.61^{\pm0.11}$ |
| Multiflow | $-7.01^{\pm1.81}$ | $0.45^{\pm0.21}$ | $0.61^{\pm0.10}$ |
| TFG-Flow | $\mathbf{-7.65}^{\pm1.89}$ | $0.47^{\pm0.19}$ | $0.64^{\pm0.10}$ |

## 4.2 STRUCTURE GUIDANCE

**Guidance target.** Following Gebauer et al. (2022), we represent the structural information of a molecule using its molecular fingerprint. This fingerprint, denoted as $c = (c_1, \ldots, c_L)$, consists of a sequence of bits that indicate the presence or absence of specific substructures within the molecule. Each substructure corresponds to a particular position $l$ in the bit vector, with the bit $c_l$ set to 1 if the substructure is present in the molecule and 0 if it is absent. To guide the generation of molecules with a desired structure (encoded by the fingerprint $c$), we define the guidance target as $f(\boldsymbol{G}) := \exp(-\|\mathcal{E}(\boldsymbol{G}) - c\|^2)$. Here, $\mathcal{E}$ refers to a multi-label classifier trained using binary cross-entropy loss to predict the molecular fingerprint, as detailed in App. D.2.

**Evaluation metrics.** We use Tanimoto coefficient (Bajusz et al., 2015) $\mathrm{TC}(c_1, c_2) = \frac{|c_1 \cap c_2|}{|c_1 \cup c_2|}$ to measure the similarity between the fingerprint $c_1$ of generated molecule and the target fingerprint $c_2$.

**Results analysis.** The results are shown in Table 3. TFG-Flow improves the similarity of unconditional generative model by 76.83% and 22.35% on QM9 and GEOM-Drug, respectively. But we still note that the Tanimoto similarity of 0.290 and 0.208 are not satisfactory for structure guidance. We will make efforts to improve training-free guidance for better performance on this task in the future.

## 4.3 POCKET-TARGETED DRUG DESIGN

We also introduce a novel benchmark for training-free guidance. In practical drug design, the goal is typically to create drugs that can bind to a specific protein target (see related work discussion in App. C), making the inclusion of pocket targets a more realistic setting for guided generation. To enhance the effectiveness of drug design, we aim for the generated molecules to exhibit strong drug-like properties, demonstrate high binding affinity to the target pocket, and be easily synthesizable. We integrate these criteria into our TFG-Flow to guide the drug design.

**Datasets and Models.** We utilize CrossDocked2020 training set to train both the unconditional flow model and the drug quality prediction network. Unlike QM9 and GEOM-Drug, the input graph $\boldsymbol{G}$ to the flow model includes not only the coordinates and atom types of molecules but also the protein pocket, which remains fixed during message passing in the EGNN. Also, there is no need to train an oracle target predictor, as the relevant metrics can be derived from publicly available chemistry software. Further details regarding the network and dataset are provided in App. D.

**Guidance target.** We use Vina score, QED score, and SA score as the proxy of binding affinity between the molecules and the protein, the drug-likeness of a molecule, and the synthetic accessibility of a molecule, respectively. We combine the three scores as a holistic evaluation of the drug quality via linear combination: $c = -0.1\times$ Vina score $+$ QED $+$ SA, and train a quality prediction network $\mathcal{E}(\boldsymbol{G})$ to approximate the value. The guidance target is given as $f(\boldsymbol{G}) := \mathcal{E}(\boldsymbol{G})$.

**Evaluation metrics.** We compute Vina Score by QVina (Alhossary et al., 2015), SA and QED by RDKit. The metrics are averaged over 100 molecules per pocket in CrossDocked test set.

**Baselines.** We compare TFG-Flow with different state-of-the-art target-aware generative models AR (Luo et al., 2021), Pocket2Mol (Peng et al., 2022), and TargetDiff (Guan et al., 2023). We implement Multiflow for target-aware molecular generation and apply TFG-Flow on it.

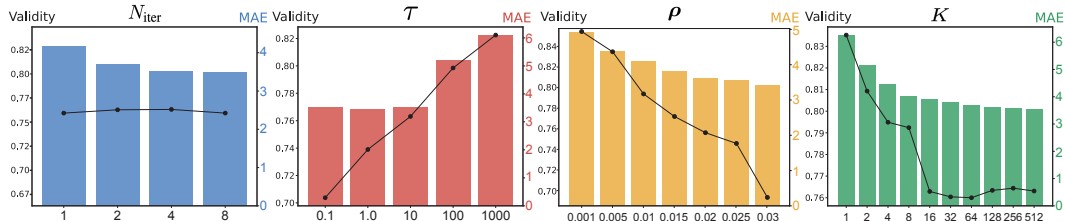

Figure 2: **Effect of varying hyperparameters** ($N_{\text{iter}}, \tau, \rho, K$)**.** The line plot indicates the validity (left $y$-axis, higher values are better), while the bar plot shows the MAE (right $y$-axis, lower values are better).

**Results Analysis.** As binding affinity is the most important metric in target-aware drug design, we prioritize Vina score in the design of our quality score. The results show that TFG-Flow improves the drug quality over unguided Multiflow and achieves the best Vina score among all the drug design methods. We also note that the autoregressive methods (AR, Pocket2Mol) generally possess better QED and SA results than diffusion-based methods (TargetDiff, Multiflow, TFG-Flow), but with weaker binding affinity. Therefore, TFG-Flow is quite effective for target-aware drug design.

### 4.4 Ablation Study

To understand how different hyper-parameters ($N_{\text{iter}}, \tau, \rho, K$) affect the performance of TFG-Flow, we conduct ablation study on quantum property guidance for polarizability $\alpha$. On this task, the searched $(\rho, \tau)$ is $(0.02, 10)$ and recall that we set $K = 512$ and $N_{\text{iter}} = 4$ constantly. In our ablation, we fixed all the other hyper-parameters and change one of them to a grid of values except for the study of $N_{\text{iter}}$, and plot the validity (the ratio of valid molecules) and guidance accuracy (MAE) in Figure 2. For $N_{\text{iter}}$, we present the results with the best $(\rho, \tau)$ for the corresponding $N_{\text{iter}}$.

The study on $N_{\text{iter}}$ shows $N_{\text{iter}} = 4$ already achieves good performance while maintaining computational efficiency, as $N_{\text{iter}} = 8$ don't bring significant improvement. For other experiments, we could see a positive correlation between validity and MAE as a trade-off between the quality of unconditional generation and the desired property alignment. Importantly, We note that the number of samples $K \approx 16$ is sufficient in our discrete guidance (Eq. (13)), which demonstrates the efficiency of our estimation technique with fast convergence rate. We can also learn that too strong guidance strength $\rho$ and $\tau$ may not improve the guidance (MAE) but will severely deteriorate the validity.

## 5 Discussions and Limitations

The related work is reviewed in App. C. Our TFG-Flow complements the trend of generative modeling through the straightforward and versatile flow matching framework (Esser et al., 2024). It also unlocks the guidance for multimodal flow (Campbell et al., 2024) and has been applied effectively to both target-agnostic and target-specific molecular design tasks. We notice that a concurrent research (Sun et al., 2024) explores training-free guidance on continuous flow in image generation, however, we have identified several theoretical concerns with this approach as detailed in App. C.2. Overall, our TFG-Flow proves to be both novel and effective, with solid theoretical foundations.

Though TFG-Flow boosts the state-of-the-art training-free guidance, a performance gap persists between training-based and training-free methods (Table 1). But as training-free guidance allows for flexible target predictor, we replace the guidance network as a pre-trained foundation model UniMol (Zhou et al., 2023) for Table 1 in App. D.6, where the performance gap with EEGSDE is further narrowed. We also notice that some literature indicates that training-free guidance tends to perform well in powerful foundation models, such as Stable Diffusion (Ye et al., 2024; Bansal et al., 2023; Yu et al., 2023). This suggests that more capable models which learn from diverse data could possibly offer better steerability. On top of that, the future trajectory for AI-Driven Drug Design might involve developing large generative foundation models and applying training-free guidance seamlessly to achieve desired properties. Beyond molecular design, our insights on multimodal guided flow are broadly applicable to other fields such as material, protein, or antibody. Given that multimodality encompasses both discrete and continuous data types, TFG-Flow provides a general framework that could handle all kinds of guidance problem. We hope that TFG-Flow will inspire further innovation within both the generative modeling and molecular design communities.

## Acknowledgement

Haowei Lin and Jianzhu Ma were supported by the National Key Plan for Scientific Research and Development of China (Grant No. 2023YFC3043300), China's Village Science and Technology City Key Technology funding, and Wuxi Research Institute of Applied Technologies, Tsinghua University (Grant No. 20242001120). Additionally, Haowei Lin and Yitao Liang received partial funding from the National Key Plan for Scientific Research and Development of China (Grant No. 2022ZD0160301).

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

# Appendix

## Table of Contents

# A  PSEUDO CODE FOR TFG-FLOW

---

**Algorithm 1** Training-free Guidance for Multimodal Flow Inference

---

**Input:** Unconditional rectified flow model $g_\theta$, target predictor $f_c$, guidance strength $\rho$, temperature $\tau$, number of steps $N_{\text{iter}}$; temporal step size $\Delta t$.

1: # Initialization
2: Sample $\boldsymbol{x}_0^{(1)}, \cdots, \boldsymbol{x}_0^{(n)} \sim$ i.i.d. $\mathcal{N}(\boldsymbol{0}, \boldsymbol{I}_3)$, $\boldsymbol{X}_0 \leftarrow \left[ \boldsymbol{x}_0^{(1)}, \cdots, \boldsymbol{x}_0^{(n)} \right]$

3: $\boldsymbol{X}_0 \leftarrow \text{Project}_\Gamma(\boldsymbol{X}_0)$; $\boldsymbol{a}_0 \leftarrow [[\text{M}], \cdots, [\text{M}]]^\top$; $t \leftarrow 0$
4: **while** $t < 1$ **do**                         ▷ Simulate Fokker-Planck and Kolmogorov Equations
5:     $\boldsymbol{X}_{1|t}, p(\boldsymbol{a}_{1|t}) \leftarrow g_\theta(\boldsymbol{X}_t, \boldsymbol{a}_t)$
6:
7:     # Discrete guidance
8:     Sample $\boldsymbol{a}_{1|t,1}, \cdots, \boldsymbol{a}_{1|t,K} \sim$ i.i.d. $p(a_{1|t})$
9:     **for** $i = 1, \cdots, n$ **do**
10:         $\hat{R}(a_t^{(i)}, b) \leftarrow \sum_{k=1}^{K} f(\boldsymbol{X}_{1|t}, \boldsymbol{a}_{1|t,k}) R_{t|1}(a_t^{(i)}, b \mid a_{1|t,k}^{(i)}) \bigg/ \sum_{k=1}^{K} f(\boldsymbol{X}_{1|t}, \boldsymbol{a}_{1|t,k})$
11:         Sample $a_{t+\Delta t}^{(i)} \sim \text{Cat}(\delta(a_{t+\Delta t}^{(i)}, a_t^{(i)}) + \hat{R}(a_t^{(i)}, a_{t+\Delta t}^{(i)}) \Delta t)$
12:     **end for**
13:
14:     # Continuous guidance
15:     Sample $\boldsymbol{a}_{1|t} \sim \text{Cat}(f_c(\boldsymbol{X}_{1|t}, \boldsymbol{a}_{1|t,k}))$          ▷ Sample $\boldsymbol{a}_{1|t}$ from $\{\boldsymbol{a}_{1|t,k}\}_{k=1}^K$ based on $f_c$
16:     **for** $j = 1, \cdots, N_{\text{iter}}$ **do**
17:         $\boldsymbol{X}_t \leftarrow \text{Project}_\Gamma \left( \boldsymbol{X}_t + \rho_t \nabla_{\boldsymbol{X}_t} \log f(g_\theta(\boldsymbol{X}_t, \boldsymbol{a}_t)_{\boldsymbol{X}}, \boldsymbol{a}_{1|t}) \right)$
18:     **end for**
19:     $\boldsymbol{X}_{1|t} \leftarrow g_\theta(\boldsymbol{X}_t, \boldsymbol{a}_t)_{\boldsymbol{X}}$
20:     **for** $i = 1, \cdots, n$ **do**
21:         $\hat{\boldsymbol{v}}(\boldsymbol{x}_t^{(i)}) \leftarrow \boldsymbol{v}_{t|1}(\boldsymbol{x}_t^{(i)} | \boldsymbol{x}_1^{(i)})$
22:         $\boldsymbol{x}_{t+\Delta t}^{(i)} \leftarrow \boldsymbol{x}_t^{(i)} + \hat{\boldsymbol{v}}(\boldsymbol{x}_t^{(i)}) \Delta t$
23:     **end for**
24:     $\boldsymbol{X}_{t+\Delta t} \leftarrow \text{Project}_\Gamma(\boldsymbol{X}_{t+\Delta t})$
25:
26:     $t = t + \Delta t$
27: **end while**
28: **Output:** $\boldsymbol{X}_1, \boldsymbol{a}_1$

---

# B  OMITTED MATHEMATICAL DERIVATIONS

In this section, we present the omitted derivation proofs for our theoretical results. Note that we always we (re)state the theorem for ease of reading, even if it has appeared in the main paper.

## B.1  FORMAL STATEMENT AND PROOF OF THEOREM 3.1

**Theorem B.1** (Existence of the guided flow (formal version of Theorem 3.1)). *Let $\mathcal{G}$ be the space of molecular representations and $\mathcal{C}$ be a finite set which includes all the values of our target property. Given a $\mathcal{G}$-valued process $\{\boldsymbol{G}_t\}_{t\in[0,1]}$ in a probability space $(\Omega_G, \mathcal{F}_G, \mathbb{P}_G)$, and a function $f_c(\boldsymbol{G}_1)$ which defines a valid distribution over $\mathcal{C}$ for any $\boldsymbol{G}_1 \in \mathcal{G}$, there exists a joint probability measure $\mathbb{P}$ on the product space $(\Omega, \mathcal{F}) = (\Omega_G \times \mathcal{C}, \mathcal{F}_G \otimes 2^{\mathcal{C}})$ and random variables $(\{\boldsymbol{G}_t\}_{t\in[0,1]}, c)$ on it that satisfies the following:*

- **Preservation of flow marginals:** *For any $0 \le t_1 < \cdots < t_m < 1$,*
$$\mathbb{P}(\boldsymbol{G}_{t_1}, \cdots, \boldsymbol{G}_{t_m}) = \mathbb{P}_G(\boldsymbol{G}_{t_1}, \cdots, \boldsymbol{G}_{t_m}).$$

- **Alignment with target predictor:** *For any $c \in \mathcal{C}$ and $\boldsymbol{G}_1 \in \mathcal{G}$, $\mathbb{P}(c|\boldsymbol{G}_1) = f_c(\boldsymbol{G}_1)$.*

- **Conditional independence of trajectory and target:** *For any $0 \le t_1 < \cdots < t_m < 1$, $(\boldsymbol{G}_{t_1}, \cdots, \boldsymbol{G}_{t_m})$ and $c$ are independent conditioning on $\boldsymbol{G}_1$.*

*Proof.* First, we construct random variables $(\{\boldsymbol{G}_t\}_{t\in[0,1]}, c)$ on the product space $(\Omega, \mathcal{F})$. For $(\omega_G, c) \in \Omega$, define $\boldsymbol{G}_t(\omega_G, c) = \boldsymbol{G}_t(\omega_G)$ ($t \in [0,1]$); $c(\omega_G, c) = c$. Note that we overload the notations $\{\boldsymbol{G}_t\}_{t\in[0,1]}$ and $c$ and redefine them as random variables in the new product space for simplicity.

Second, we construction of the joint probability measure $\mathbb{P}$ on the product space $(\Omega, \mathcal{F})$.

For each $\boldsymbol{G}_1' \in \mathcal{G}$, define a probability measure $\mathbb{P}_G^{\boldsymbol{G}_1'}$ on $\Omega_G$ by

$$\mathbb{P}_G^{\boldsymbol{G}_1'}(E_G) = \mathbb{P}_G\left(E_G \mid \boldsymbol{G}_1 = \boldsymbol{G}_1'\right) \quad \text{for any } E_G \in \mathcal{F}_G. \tag{17}$$

Again, for each $\boldsymbol{G}_1' \in \mathcal{G}$, define a probability measure $\mathbb{P}_C^{\boldsymbol{G}_1'}$ on $\mathcal{C}$ by $\mathbb{P}_C^{\boldsymbol{G}_1'}(c') = f_{c'}(\boldsymbol{G}_1')$. By definition of $f_c$, $\mathbb{P}_C^{\boldsymbol{G}_1}$ is a valid distribution.

Then, we can define $\mathbb{P}^{\boldsymbol{G}_1'}$ on $\Omega$ as

$$\mathbb{P}^{\boldsymbol{G}_1'}(E_G \times E_C) = \mathbb{P}_G^{\boldsymbol{G}_1'}(E_G)\mathbb{P}_C^{\boldsymbol{G}_1'}(E_C) \quad \text{for any } E_G \in \mathcal{F}_G; \ E_C \in 2^{\mathcal{C}}. \tag{18}$$

We obtain the joint probability measure $\mathbb{P}$ on the product space $(\Omega, \mathcal{F})$ by integrating over $\boldsymbol{G}_1' \in \mathcal{G}$:

$$\mathbb{P}(E) = \int_{\boldsymbol{G}_1' \in \mathcal{G}} \mathbb{P}^{\boldsymbol{G}_1'}(E)\mathbb{P}_{G,1}(\mathrm{d}\boldsymbol{G}_1') \quad \text{for any } E \in \mathcal{F}, \tag{19}$$

where $\mathbb{P}_{G,1}(\cdot)$ is the marginal distribution (law) of $\boldsymbol{G}_1$ in $(\Omega_G, \mathcal{F}_G, \mathbb{P}_G)$.

Now we are ready to verify that the above joint probability measure $\mathbb{P}$ satisfies the desired properties.

**Preservation of flow marginals.** For any $E_G \in \mathcal{F}_G$,

$$\mathbb{P}(E_G \times \mathcal{C}) = \int_{\mathcal{G}} \mathbb{P}^{\boldsymbol{G}_1'}(E_G \times \mathcal{C})\mathbb{P}_{G,1}(\mathrm{d}\boldsymbol{G}_1') = \int_{\mathcal{G}} \mathbb{P}_G^{\boldsymbol{G}_1'}(E_G)\mathbb{P}_{G,1}(\mathrm{d}\boldsymbol{G}_1') = \mathbb{P}_G(E_G). \tag{20}$$

Thus, the marginal distribution of $\{\boldsymbol{G}_t\}_{t\in[0,1]}$ under $\mathbb{P}$ is $\mathbb{P}_G$.

Specifically, for any $0 \le t_1 < \cdots < t_m < 1$,

$$\mathbb{P}(\boldsymbol{G}_{t_1}, \cdots, \boldsymbol{G}_{t_m}) = \mathbb{P}_G(\boldsymbol{G}_{t_1}, \cdots, \boldsymbol{G}_{t_m}).$$

**Alignment with Target Predictor.** Note that in our definition, $\mathbb{P}^{\boldsymbol{G}_1'}(\cdot) = \mathbb{P}(\cdot \mid \boldsymbol{G}_1 = \boldsymbol{G}_1')$. Therefore, for any $\boldsymbol{G}_1' \in \mathcal{G}$ and $c' \in \mathcal{C}$, we have

$$\mathbb{P}(c = c' \mid \boldsymbol{G}_1 = \boldsymbol{G}_1') = \mathbb{P}^{\boldsymbol{G}_1'}(\Omega_{\mathrm{G}} \times \{c'\}) = \mathbb{P}_{\mathrm{G}}^{\boldsymbol{G}_1'}(\Omega_{\mathrm{G}})\mathbb{P}_{\mathrm{C}}^{\boldsymbol{G}_1'}(\{c'\}) = \mathbb{P}_{\mathrm{C}}^{\boldsymbol{G}_1'}(\{c'\}) = f_c(\boldsymbol{G}_1'). \quad (21)$$

This establishes the specified alignment with the target predictor.

**Conditional independence of trajectory and target.** Let $0 \le t_1 < t_2 < \cdots < t_m < 1$, and let $\varphi_{\mathrm{G}} : \mathcal{G}^m \to \mathbb{R}$ and $\varphi_{\mathrm{C}} : \mathcal{C} \to \mathbb{R}$ be any bounded measurable functions. We need to show that

$$\mathbb{E}_{\mathbb{P}}\left[\varphi_{\mathrm{G}}(\boldsymbol{G}_{t_1}, \cdots, \boldsymbol{G}_{t_m})\varphi_{\mathrm{C}}(c) \mid \boldsymbol{G}_1\right] = \mathbb{E}_{\mathbb{P}}\left[\varphi_{\mathrm{G}}(\boldsymbol{G}_{t_1}, \cdots, \boldsymbol{G}_{t_m}) \mid \boldsymbol{G}_1\right] \mathbb{E}_{\mathbb{P}}\left[\varphi_{\mathrm{C}}(c) \mid \boldsymbol{G}_1\right].$$

Note that under $\mathbb{P}_{\mathrm{G}}$, given $\boldsymbol{G}_1$, the random variables $(\boldsymbol{G}_{t_1}, \cdots, \boldsymbol{G}_{t_m})$ depend only on $\omega_{\mathrm{G}}$, and $c$ depends on $\omega_{\mathrm{G}}$ only through $\boldsymbol{G}_1$. Therefore,

$$\mathbb{E}_{\mathbb{P}}\left[\varphi_{\mathrm{G}}(\boldsymbol{G}_{t_1}, \cdots, \boldsymbol{G}_{t_m})\varphi_{\mathrm{C}}(c) \mid \boldsymbol{G}_1\right] \tag{22}$$

$$= \int_{\Omega_{\mathrm{G}} \times \mathcal{C}} \varphi_{\mathrm{G}}\left(\boldsymbol{G}_{t_1}(\omega_{\mathrm{G}}', c'), \cdots, \boldsymbol{G}_{t_m}(\omega_{\mathrm{G}}', c')\right) \varphi_{\mathrm{C}}\left(\omega_{\mathrm{G}}', c'\right) \mathbb{P}^{\boldsymbol{G}_1}\left(\mathrm{d}\omega_{\mathrm{G}}' \mathrm{d}c'\right) \tag{23}$$

$$= \int_{\Omega_{\mathrm{G}} \times \mathcal{C}} \varphi_{\mathrm{G}}\left(\boldsymbol{G}_{t_1}(\omega_{\mathrm{G}}', c'), \cdots, \boldsymbol{G}_{t_m}(\omega_{\mathrm{G}}', c')\right) \varphi_{\mathrm{C}}\left(\omega_{\mathrm{G}}', c'\right) \mathbb{P}_{\mathrm{G}}^{\boldsymbol{G}_1}\left(\mathrm{d}\omega_{\mathrm{G}}'\right) \mathbb{P}_{\mathrm{C}}^{\boldsymbol{G}_1}\left(\mathrm{d}c'\right) \tag{24}$$

$$= \left(\int_{\Omega_{\mathrm{G}}} \varphi_{\mathrm{G}}(\boldsymbol{G}_{t_1}(\omega_{\mathrm{G}}'), \cdots, \boldsymbol{G}_{t_m}(\omega_{\mathrm{G}'})) \mathbb{P}_{\mathrm{G}}^{\boldsymbol{G}_1}\left(\mathrm{d}\omega_{\mathrm{G}}'\right)\right) \cdot \left(\int_{\mathcal{C}} \varphi_{\mathrm{C}}(c') \mathbb{P}_{\mathrm{C}}^{\boldsymbol{G}_1}\left(\mathrm{d}c'\right)\right) \tag{25}$$

$$= \mathbb{E}_{\mathbb{P}_{\mathrm{G}}^{\boldsymbol{G}_1}}\left[\varphi_{\mathrm{G}}(\boldsymbol{G}_{t_1}, \cdots, \boldsymbol{G}_{t_m})\right] \cdot \mathbb{E}_{\mathbb{P}_{\mathrm{C}}^{\boldsymbol{G}_1}}\left[\varphi_{\mathrm{C}}(c)\right] \tag{26}$$

$$= \mathbb{E}_{\mathbb{P}}\left[\varphi_{\mathrm{G}}(\boldsymbol{G}_{t_1}, \cdots, \boldsymbol{G}_{t_m}) \mid \boldsymbol{G}_1\right] \cdot \mathbb{E}_{\mathbb{P}}\left[\varphi_{\mathrm{C}}(c) \mid \boldsymbol{G}_1\right], \tag{27}$$

indicating that $(\boldsymbol{G}_{t_1}, \cdots, \boldsymbol{G}_{t_m})$ and $c$ are independent conditioning on $\boldsymbol{G}_1$.

To sum up, we have constructed a probability measure $\mathbb{P}$ on $(\Omega, \mathcal{F})$ and verify that $\mathbb{P}$ satisfies the desired properties, which concludes the proof. $\qquad \square$

**Remark.** It's easy to see that our proof also applies when $\mathcal{C} = \mathbb{R}$. In this case, the joint probability measure $\mathbb{P}$ needs to be defined on $(\Omega, \mathcal{F}) = (\Omega_{\mathrm{G}} \times \mathbb{R}, \mathcal{F}_{\mathrm{G}} \otimes \mathcal{B}(\mathbb{R}))$, where $\mathcal{B}(\mathbb{R})$ denotes the Borel $\sigma$-algebra of $\mathbb{R}$. This lays the foundation of our method in the setting where $f_c$ is a regression model.

## B.2 FORMAL STATEMENT AND PROOF OF THEOREM 3.2

We present the formal version of Theorem 3.2 as two separate theorems: Theorem B.2 for the continuous flow and Theorem B.3 for the discrete flow.

**Theorem B.2** (Guided velocity). *Let $\{p_t(\boldsymbol{x}_t)\}_{t \in [0,1]}$ be a continuous flow on $\mathbb{R}^3$. Let $\boldsymbol{v}_{t|1}(\boldsymbol{x}_t|\boldsymbol{x}_1)$ be the conditional velocity that generates $p_{t|1}(\boldsymbol{x}_t|\boldsymbol{x}_1)$ via Fokker-Planck Equation, i.e., $\partial_t p_{t|1} = -\nabla \cdot (\boldsymbol{v}_{t|1} p_{t|1})$. Then the guided flow $\{p_t(\boldsymbol{x}_t|c)\}_{t \in [0,1]}$ defined via the construction of Theorem 3.1 can be generated by the following guided velocity via Fokker-Planck Equation:*

$$\boldsymbol{v}_t(\boldsymbol{x}_t|c) = \mathbb{E}_{p_{1|t}(\boldsymbol{x}_1|\boldsymbol{x}_t,c)} \left[ \boldsymbol{v}_{t|1}(\boldsymbol{x}_t|\boldsymbol{x}_1) \right]. \tag{28}$$

*Proof.* We begin with the Fokker-Planck Equation of $p_{t|1}(\boldsymbol{x}_t|\boldsymbol{x}_1)$:

$$\partial_t p_{t|1}(\boldsymbol{x}_t|\boldsymbol{x}_1) = -\nabla \cdot \left( \boldsymbol{v}_{t|1}(\boldsymbol{x}_t|\boldsymbol{x}_1) p_{t|1}(\boldsymbol{x}_t|\boldsymbol{x}_1) \right). \tag{29}$$

Taking expectation with respect to $p_{\text{data}}(\boldsymbol{x}_1|c)$ over both sides yields

$$\mathbb{E}_{p_{\text{data}}(\boldsymbol{x}_1|c)} \left[ \partial_t p_{t|1}(\boldsymbol{x}_t|\boldsymbol{x}_1) \right] = -\mathbb{E}_{p_{\text{data}}(\boldsymbol{x}_1|c)} \left[ \nabla \cdot \left( \boldsymbol{v}_{t|1}(\boldsymbol{x}_t|\boldsymbol{x}_1) p_{t|1}(\boldsymbol{x}_t|\boldsymbol{x}_1) \right) \right]. \tag{30}$$

The left-hand size of Eq. (30) can be simplified as

$$\mathbb{E}_{p_{\text{data}}(\boldsymbol{x}_1|c)} \left[ \partial_t p_{t|1}(\boldsymbol{x}_t|\boldsymbol{x}_1) \right] = \int_{\mathbb{R}^3} p_{\text{data}}(\boldsymbol{x}_1|c) \partial_t p_{t|1}(\boldsymbol{x}_t|\boldsymbol{x}_1) \mathrm{d}\boldsymbol{x}_1 \tag{31}$$

$$= \partial_t \left( \int_{\mathbb{R}^3} p_{\text{data}}(\boldsymbol{x}_1|c) p_{t|1}(\boldsymbol{x}_t|\boldsymbol{x}_1) \mathrm{d}\boldsymbol{x}_1 \right) \tag{32}$$

$$= \partial_t \left( \int_{\mathbb{R}^3} p_{\text{data}}(\boldsymbol{x}_1|c) p_{t|1}(\boldsymbol{x}_t|\boldsymbol{x}_1, c) \mathrm{d}\boldsymbol{x}_1 \right) \tag{33}$$

$$= \partial_t p_t(\boldsymbol{x}_t|c) \tag{34}$$

Note that in Eq. (33), we use the conditional independence property of trajectory and target and get $p_{\text{data}}(\boldsymbol{x}_1|c) p_{t|1}(\boldsymbol{x}_t|\boldsymbol{x}_1) = p_t(\boldsymbol{x}_t, \boldsymbol{x}_1|c)$.

For the right-hand size of Eq. (30), we have

$$- \mathbb{E}_{p_{\text{data}}(\boldsymbol{x}_1|c)} \left[ \nabla \cdot \left( \boldsymbol{v}_{t|1}(\boldsymbol{x}_t|\boldsymbol{x}_1) p_{t|1}(\boldsymbol{x}_t|\boldsymbol{x}_1) \right) \right] \tag{35}$$

$$= - \int_{\mathbb{R}^3} \nabla \cdot \left( \boldsymbol{v}_{t|1}(\boldsymbol{x}_t|\boldsymbol{x}_1) p_{t|1}(\boldsymbol{x}_t|\boldsymbol{x}_1) p_{\text{data}}(\boldsymbol{x}_1|c) \mathrm{d}\boldsymbol{x}_1 \right) \tag{36}$$

$$= - \nabla \cdot \left( \int_{\mathbb{R}^3} \boldsymbol{v}_{t|1}(\boldsymbol{x}_t|\boldsymbol{x}_1) p_{t|1}(\boldsymbol{x}_t|\boldsymbol{x}_1) p_{\text{data}}(\boldsymbol{x}_1|c) \mathrm{d}\boldsymbol{x}_1 \right) \tag{37}$$

$$= - \nabla \cdot \left( \int_{\mathbb{R}^3} \boldsymbol{v}_{t|1}(\boldsymbol{x}_t|\boldsymbol{x}_1) p_t(\boldsymbol{x}_t, \boldsymbol{x}_1|c) \mathrm{d}\boldsymbol{x}_1 \right) \tag{38}$$

$$= - \nabla \cdot \left( p_t(\boldsymbol{x}_t|c) \int_{\mathbb{R}^3} \boldsymbol{v}_{t|1}(\boldsymbol{x}_t|\boldsymbol{x}_1) p_t(\boldsymbol{x}_1|\boldsymbol{x}_t, c) \mathrm{d}\boldsymbol{x}_1 \right) \tag{39}$$

$$= - \nabla \cdot \left( \boldsymbol{v}_t(\boldsymbol{x}_t|c) p_t(\boldsymbol{x}_t|c) \right) \tag{40}$$

Putting the above together leads to $\partial_t p_t(\boldsymbol{x}_t|c) = -\nabla \cdot (\boldsymbol{v}_t(\boldsymbol{x}_t|c) p_t(\boldsymbol{x}_t|c))$. □

**Theorem B.3** (Guided rate matrix). *Let $\{p_t(a_t)\}_{t \in [0,1]}$ be a discrete flow on $\mathcal{A}$. Let $\boldsymbol{R}_{t|1}(a_t, b|a_1)$ be the conditional velocity that generates $\boldsymbol{p}_{t|1} = (p_{t|1}(a|a_1))_{a \in \mathcal{A}}$ via Kolmogorov Equation, i.e., $\partial_t \boldsymbol{p}_{t|1} = \boldsymbol{p}_{t|1} \boldsymbol{R}_{t|1}$. Then the guided flow $\{p_t(a_t|c)\}_{t \in [0,1]}$ defined via the construction of Theorem 3.1 can be generated by the following guided rate matrix via Kolmogorov Equation:*

$$R_t(a_t, b|c) = \mathbb{E}_{p_{1|t}(a_1|a_t,c)} \left[ R_{t|1}(a_t, b|a_1) \right]. \tag{41}$$

*Proof.* The proof idea is exactly the same as that of Theorem B.2, i.e., taking expectation with respect to $p_{\text{data}}(a_1|c)$ over both sides of Kolmogorov Equation and simplifying them. Again, $p_{\text{data}}(a_1|c) p_{t|1}(a_t|a_1) = p_t(a_t, a_1|c)$ will be useful in the simplification. □

### B.3 IMPORTANCE SAMPLING FAILS FOR SAMPLING THE DISCRETE GUIDED FLOW

**Proposition B.4** (Naive importance sampling for the discrete guided flow). *For $t \in [0, 1)$, the guided rate matrix defined in Theorem 3.2 satisfies*

$$R_t(a_t^{(i)}, j|c) = \mathbb{E}_{p_{1|t}(\boldsymbol{G}_1|\boldsymbol{G}_t)} \left[ \frac{p_c(c|\boldsymbol{G}_1)}{p(c|\boldsymbol{G}_t)} R_{t|1}(a_t^{(i)}, j|a_1^{(i)}) \right] \tag{42}$$

*Proof.* By definition of the guided rate matrix and importance sampling, we have

$$R_t(a_t^{(i)}, j|c) = \mathbb{E}_{p_{1|t}(\boldsymbol{G}_1|\boldsymbol{G}_t)} \left[ \frac{p_{1|t}(\boldsymbol{G}_1|\boldsymbol{G}_t, c)}{p_{1|t}(\boldsymbol{G}_1|\boldsymbol{G}_t)} R_{t|1}(a_t^{(i)}, j|a_1^{(i)}) \right]. \tag{43}$$

Note that for any $t \in [0, 1)$, $\boldsymbol{G}_t$ and $c$ are conditionally independent given $\boldsymbol{G}_1$. Therefore,

$$p_{1|t}(\boldsymbol{G}_1|\boldsymbol{G}_t, c) = \frac{p(\boldsymbol{G}_1, c|\boldsymbol{G}_t)}{p(c|\boldsymbol{G}_t)} = \frac{p_{1|t}(\boldsymbol{G}_1|\boldsymbol{G}_t)p(c|\boldsymbol{G}_1, \boldsymbol{G}_t)}{p(c|\boldsymbol{G}_t)} = \frac{p_{1|t}(\boldsymbol{G}_1|\boldsymbol{G}_t)p(c|\boldsymbol{G}_1)}{p(c|\boldsymbol{G}_t)} \tag{44}$$

$$\Rightarrow \frac{p_{1|t}(\boldsymbol{G}_1|\boldsymbol{G}_t, c)}{p_{1|t}(\boldsymbol{G}_1|\boldsymbol{G}_t)} = \frac{p(c|\boldsymbol{G}_1)}{p(c|\boldsymbol{G}_t)}, \tag{45}$$

which concludes the proof. $\square$

**Proposition B.5** (Restatement of Proposition 3.3). *For $t \in [0, 1)$, the guided rate matrix defined in Theorem 3.2 satisfies*

$$R_t(a_t^{(i)}, j|c) = \frac{\mathbb{E}_{p_{1|t}(\boldsymbol{G}_1|\boldsymbol{G}_t)} \left[ f_c(\boldsymbol{G}_1) R_{t|1}(a_t^{(i)}, j|a_1^{(i)}) \right]}{\mathbb{E}_{p_{1|t}(\boldsymbol{G}_1|\boldsymbol{G}_t)} \left[ f_c(\boldsymbol{G}_1) \right]}. \tag{46}$$

*Proof.* Proposition B.4 indicates that

$$R_t(a_t^{(i)}, j|c) = \frac{\mathbb{E}_{p_{1|t}(\boldsymbol{G}_1|\boldsymbol{G}_t)} \left[ f_c(\boldsymbol{G}_1) R_{t|1}(a_t^{(i)}, j|a_1^{(i)}) \right]}{p(c|\boldsymbol{G}_t)} \tag{47}$$

By Conditional independence of trajectory and target in the guided flow, we have

$$p(c|\boldsymbol{G}_t) = \int_{\mathcal{G}} p(c|\boldsymbol{G}_1, \boldsymbol{G}_t) p_{1|t}(\boldsymbol{G}_1|\boldsymbol{G}_t) \mathrm{d}\boldsymbol{G}_1 \tag{48}$$

$$= \int_{\mathcal{G}} p(c|\boldsymbol{G}_1) p_{1|t}(\boldsymbol{G}_1|\boldsymbol{G}_t) \mathrm{d}\boldsymbol{G}_1 \tag{49}$$

$$= \mathbb{E}_{p_{1|t}(\boldsymbol{G}_1|\boldsymbol{G}_t)} \left[ f_c(\boldsymbol{G}_1) \right] \tag{50}$$

Combining Eq. (47) and Eq. (50) leads to the conclusion. $\square$

### B.4 PROOF OF THEOREM 3.4

**Theorem B.6** (Restatement of Theorem 3.4). *Let $\boldsymbol{G}_{1|t,1}, \cdots, \boldsymbol{G}_{1|t,K} \sim$ i.i.d. $p_{1|t}(\cdot|\boldsymbol{G}_t)$. Define the estimation of $R_t(a_t^{(i)}, b|c)$ as*

$$\hat{R}_t(a_t^{(i)}, b|c) = \sum_{k=1}^{K} f_c(\boldsymbol{G}_{1|t,k}) R_{t|1}(a_t^{(i)}, b \mid a_{1|t,k}^{(i)}) \Big/ \sum_{k=1}^{K} f_c(\boldsymbol{G}_{1|t,k}). \tag{51}$$

*Assume $\underline{f_c} = \inf\limits_{\boldsymbol{G} \in \mathcal{G}} f_c(\boldsymbol{G}) > 0$. Given any $\varepsilon \in (0, \underline{f_c}/2)$, $\delta \in (0,1)$, if $K = \Theta\left(\frac{1}{\varepsilon^2} \log \frac{n|\mathcal{A}|}{\delta}\right)$, then*

$$\mathbb{P}\left(\sup_{i \in \{1, \cdots, n\},\ b \in \mathcal{A}} \left| \hat{R}_t(a_t^{(i)}, b|c) - R_t(a_t^{(i)}, b|c) \right| < \varepsilon\right) \geq 1 - \delta \tag{52}$$

*Proof.* Consider some fixed $i \in \{1, \cdots, n\}$, $b \in \mathcal{A}$. To simplify our exposition, denote

$$N_{i,b} = \mathbb{E}_{p_{1|t}(\boldsymbol{G}_1|\boldsymbol{G}_t)} \left[ f_c(\boldsymbol{G}_1) R_{t|1}(a_t^{(i)}, j|a_1^{(i)}) \right] \tag{53}$$

$$D_{i,b} = \mathbb{E}_{p_{1|t}(\boldsymbol{G}_1|\boldsymbol{G}_t)} \left[ f_c(\boldsymbol{G}_1) \right] \tag{54}$$

$$\hat{N}_{i,b} = \frac{1}{K} \sum_{k=1}^{K} f_c(\boldsymbol{G}_{1|t,k}) R_{t|1}(a_t^{(i)}, b \mid a_{1|t,k}^{(i)}) \tag{55}$$

$$\hat{D}_{i,b} = \frac{1}{K} \sum_{k=1}^{K} f_c(\boldsymbol{G}_{1|t,k}) \tag{56}$$

We note that for any $k \in \{1, \cdots, K\}$,

$$0 < f_c(\boldsymbol{G}_{1|t,k}) R_{t|1}(a_t^{(i)}, b \mid a_{1|t,k}^{(i)}) < \frac{1}{1-t}. \tag{57}$$

Therefore, by Hoeffding's inequality, we have

$$\mathbb{P}\left(\left| N_{i,b} - \hat{N}_{i,b} \right| > \frac{\varepsilon(1-t)\underline{f_c}^2}{4}\right) \leq 2 \exp\left(-\frac{K\varepsilon^2(1-t)^4\underline{f_c}^4}{8}\right). \tag{58}$$

Similarly, for $\left| D_{i,b} - \hat{D}_{i,b} \right|$, we have

$$\mathbb{P}\left(\left| D_{i,b} - \hat{D}_{i,b} \right| > \frac{\varepsilon(1-t)\underline{f_c}^2}{4}\right) \leq 2 \exp\left(-\frac{K\varepsilon^2(1-t)^4\underline{f_c}^4}{8}\right). \tag{59}$$

Suppose $\left| N_{i,b} - \hat{N}_{i,b} \right| \leq \frac{\varepsilon(1-t)\underline{f_c}^2}{4}$ and $\left| D_{i,b} - \hat{D}_{i,b} \right| \leq \frac{\varepsilon(1-t)\underline{f_c}^2}{4}$, then we have

$$\left| \frac{N_{i,b}}{D_{i,b}} - \frac{\hat{N}_{i,b}}{\hat{D}_{i,b}} \right| = \frac{\left| N_{i,b}\hat{D}_{i,b} - \hat{N}_{i,b}D_{i,b} \right|}{D_{i,b}\hat{D}_{i,b}} \tag{60}$$

$$\leq \frac{D_{i,b}\left| N_{i,b} - \hat{N}_{i,b} \right| + N_{i,b}\left| \hat{D}_{i,b} - D_{i,b} \right|}{D_{i,b}\hat{D}_{i,b}} \tag{61}$$

$$\leq \frac{D_{i,b} + N_{i,b}}{D_{i,b}\hat{D}_{i,b}} \cdot \frac{\varepsilon(1-t)\underline{f_c}^2}{4} \tag{62}$$

$$\leq \frac{\varepsilon\underline{f_c}^2}{2D_{i,b}\hat{D}_{i,b}} \qquad \left(\text{Note that } 0 \leq D_{i,b}, N_{i,b} \leq \frac{1}{1-t}\right) \tag{63}$$

Recall the definition of $\underline{f_c}$ and the fact that $\varepsilon \in \left(0, \underline{f_c}/2\right)$, we have $D_{i,b} \geq \underline{f_c}$ and $D_{i,b} > \underline{f_c}/2$. Plugging these two lower bounds into Eq. (63) yields $\left| \dfrac{N_{i,b}}{D_{i,b}} - \dfrac{\hat{N}_{i,b}}{\hat{D}_{i,b}} \right| < \varepsilon.$

Thus $\left| \dfrac{N_{i,b}}{D_{i,b}} - \dfrac{\hat{N}_{i,b}}{\hat{D}_{i,b}} \right| \geq \varepsilon$ must imply $\left| N_{i,b} - \hat{N}_{i,b} \right| > \dfrac{\varepsilon(1-t)\underline{f_c}^2}{4}$ or $\left| D_{i,b} - \hat{D}_{i,b} \right| > \dfrac{\varepsilon(1-t)\underline{f_c}^2}{4}.$

$$\mathbb{P}\left( \left| \hat{R}_t(a_t^{(i)}, b|c) - R_t(a_t^{(i)}, b|c) \right| \geq \varepsilon \right) \tag{64}$$

$$= \mathbb{P}\left( \left| \dfrac{N_{i,b}}{D_{i,b}} - \dfrac{\hat{N}_{i,b}}{\hat{D}_{i,b}} \right| \geq \varepsilon \right) \tag{65}$$

$$\leq \mathbb{P}\left( \left| N_{i,b} - \hat{N}_{i,b} \right| > \dfrac{\varepsilon(1-t)\underline{f_c}^2}{4} \right) + \mathbb{P}\left( \left| D_{i,b} - \hat{D}_{i,b} \right| > \dfrac{\varepsilon(1-t)\underline{f_c}^2}{4} \right) \tag{66}$$

$$\leq 4\exp\left( -\dfrac{K\varepsilon^2(1-t)^4\underline{f_c}^4}{8} \right). \tag{67}$$

Applying union bound on $i \in \{1, \cdots, n\}$, $b \in \mathcal{A}$ yields

$$\mathbb{P}\left( \sup_{i \in \{1,\cdots,n\},\, b \in \mathcal{A}} \left| \hat{R}_t(a_t^{(i)}, b|c) - R_t(a_t^{(i)}, b|c) \right| \geq \varepsilon \right) \leq 4n|\mathcal{A}|\exp\left( -\dfrac{K\varepsilon^2(1-t)^4\underline{f_c}^4}{8} \right).$$

Set

$$K = \dfrac{8}{\varepsilon^2(1-t)^4\underline{f_c}^4} \log \dfrac{4n|\mathcal{A}|}{\delta}. \tag{68}$$

Then we have

$$\mathbb{P}\left( \sup_{i \in \{1,\cdots,n\},\, b \in \mathcal{A}} \left| \hat{R}_t(a_t^{(i)}, b|c) - R_t(a_t^{(i)}, b|c) \right| \geq \varepsilon \right) \leq \delta, \tag{69}$$

which concludes the proof. $\qquad\square$

## B.5 FORMAL STATEMENT AND PROOF OF THEOREM 3.5

Before stating and proving the formal version of Theorem 3.5, we present additional definitions and technical lemmas:

**Definition 1** (SO(3)-invariance). *We say a mapping $h : \mathbb{R}^{3 \times n} \to \mathbb{R}$ is SO(3)-**invariant** if*

$$h(\boldsymbol{S}\boldsymbol{X}) = h(\boldsymbol{X}) \qquad \text{for any } \boldsymbol{X} \in \mathbb{R}^{3 \times n} \text{ and } \boldsymbol{S} \in \text{SO}(3). \tag{70}$$

**Definition 2** (SO(3)-equivariance). *We say a mapping $h : \mathbb{R}^{3 \times n} \to \mathbb{R}^{3 \times n}$ is SO(3)-**equivariant** if*

$$h(\boldsymbol{S}\boldsymbol{X}) = \boldsymbol{S}h(\boldsymbol{X}) \qquad \text{for any } \boldsymbol{X} \in \mathbb{R}^{3 \times n} \text{ and } \boldsymbol{S} \in \text{SO}(3). \tag{71}$$

**Lemma B.7** (Gradient of invariant mappings). *Let $h : \mathbb{R}^{3 \times n} \to \mathbb{R}$ be an SO(3)-invariant mapping. Then the gradient $\nabla h$ is an SO(3)-equivariant mapping.*

*Proof.* By chain rule and SO(3)-invariance, we have $\nabla h(\boldsymbol{S}\boldsymbol{X}) = \boldsymbol{S}\nabla h(\boldsymbol{X})$ for any $\boldsymbol{S} \in \text{SO}(3)$. □

**Lemma B.8** (Composition and sum of equivariant mappings). *Let $h_1$ and $h_2$ be SO(3)-equivariant mappings. Then their composition $h_2 \circ h_1$ and their sum $h_1 + h_2$ is also SO(3)-equivariant.*

*Proof.* The lemma follows immediately from the definition of SO(3)-equivariant mappings. □

**Lemma B.9** (Push-forward of equivariant mappings). *Let $h$ be an SO(3)-equivariant mapping and $p$ be an SO(3)-invariant distribution. Then $h_\# p$ is also an SO(3)-invariant distribution, where $\#$ denotes the push-forward operator.*

*Proof.* The lemma follows immediately from the definition of SO(3)-equivariant mappings and SO(3)-invariant distributions. □

**Theorem B.10** (SO(3)-invariance (formal version of Theorem 3.5)). *Consider Algorithm 1. Assume the target predictor $f_c(\boldsymbol{G})$ is SO(3)-invariant, the flow model $g_\theta(\boldsymbol{G})$ is SO(3)-equivariant, and the distribution of $\boldsymbol{G}_0$ is SO(3)-invariant w.r.t. the atomic coordinates $\boldsymbol{X}$. Let $W = 1/\Delta t$. Then for any $w \in \{0, \cdots, W\}$, the distribution of $\boldsymbol{X}_{w\Delta t}$ is SO(3)-invariant.*

*Proof.* We prove by induction on $w$.

If $w = 0$, then the distribution of $\boldsymbol{X}_{w\Delta t}$ is SO(3)-invariant according to the assumption.

Assume that the distribution of $\boldsymbol{X}_{(w-1)\Delta t}$ is SO(3)-invariant for some $w \in \{1, \cdots, W\}$. We show that the distribution of $\boldsymbol{X}_{w\Delta t}$ is SO(3)-invariant.

We note that in Algorithm 1, $\boldsymbol{X}_{w\Delta t}$ is generated from $\boldsymbol{X}_{(w-1)\Delta t}$ via a series of mappings: $\boldsymbol{X}_{w\Delta t} = h_L \circ \cdots \circ h_1(\boldsymbol{X}_{(w-1)\Delta t})$, where each $h_i$ is one of the following:

- Projection: $\boldsymbol{X} \mapsto \text{Project}_\Gamma(\boldsymbol{X})$.

- Guidance based on the target predictor: $\boldsymbol{X} \mapsto \boldsymbol{X} + \rho_t \nabla_{\boldsymbol{X}} \log f_c(g_\theta(\boldsymbol{X}, \boldsymbol{a}_t)_{\boldsymbol{X}}, \boldsymbol{a}_{1|t})$.

- Euler step: $\left[\boldsymbol{x}^{(i)}\right]_{i=1}^n \mapsto \left[\boldsymbol{x}^{(i)} + \boldsymbol{v}_{t|1}(\boldsymbol{x}^{(i)}|\boldsymbol{x}_1^{(i)})\Delta t\right]$.

We show that all these mappings are SO(3)-equivariant.

**Projection.**  By definition of $\text{Project}_\Gamma$ (in Sec. 2.1), for any $\boldsymbol{S} \in \text{SO}(3)$,

$$\text{Project}_\Gamma(\boldsymbol{S}\boldsymbol{X}) = \boldsymbol{S}\boldsymbol{X} - \frac{1}{n}\sum_{i=1}^n \boldsymbol{S}\boldsymbol{x}^{(i)}\mathbf{1}_n^\top = \boldsymbol{S}\left(\boldsymbol{X} - \frac{1}{n}\sum_{i=1}^n \boldsymbol{x}^{(i)}\mathbf{1}_n^\top\right) = \boldsymbol{S}\text{Project}_\Gamma(\boldsymbol{X}). \tag{72}$$

**Guidance based on the target predictor.**  According to the assumption, $\boldsymbol{X} \mapsto \log f_c(\boldsymbol{X}, \boldsymbol{a}_{1|t})$ is SO(3)-invariant. Thus, by lemma B.7, its gradient is SO(3)-equivariant. Also note that $\boldsymbol{X} \mapsto g_\theta(\boldsymbol{X}, \boldsymbol{a}_t)_{\boldsymbol{X}}$ is SO(3)-invariant. Thus, by lemma B.8, $\boldsymbol{X} \mapsto \boldsymbol{X} + \rho_t \nabla_{\boldsymbol{X}} \log f_c(g_\theta(\boldsymbol{X}, \boldsymbol{a}_t)_{\boldsymbol{X}}, \boldsymbol{a}_{1|t})$ is SO(3)-equivariant.

**Euler step.** By definition of $\boldsymbol{v}_{t|1}(\boldsymbol{x}^{(i)}|\boldsymbol{x}_1^{(i)})$ (in Eq. (9)), we can easily check the SO(3)-equivariance of $\left[\boldsymbol{x}^{(i)}\right]_{i=1}^n \mapsto \left[\boldsymbol{x}^{(i)} + \boldsymbol{v}_{t|1}(\boldsymbol{x}^{(i)}|\boldsymbol{x}_1^{(i)})\Delta t\right]$.

To sum up, $h_1, \cdots, h_L$ are all SO(3)-equivariant. By lemma B.8, $h_L \circ \cdots \circ h_1$ is SO(3)-equivariant.

Recall that the distribution of $\boldsymbol{X}_{(w-1)\Delta t}$ is SO(3)-invariant. Therefore, by lemma B.9, the distribution of $\boldsymbol{X}_{w\Delta t}$ is also SO(3)-invariant. We conclude the proof by induction. $\qquad\square$

# C DISCUSSION OF RELATED WORK

## C.1 GENERAL DISCUSSION

**Diffusion and Flow Matching** Diffusion models (Sohl-Dickstein et al., 2015; Song & Ermon, 2019; Song et al., 2020; 2021; Dhariwal & Nichol, 2021; Song & Ermon, 2020; Karras et al., 2022; Hoogeboom et al., 2023; Han et al., 2022b; Qin et al., 2023b) have demonstrated exceptional performance across a range of generative modeling tasks, including image and video generation (Saharia et al., 2022; Ho et al., 2022; Zhang et al., 2024), audio generation (Kong et al., 2020), and 3D geometry generation (Luo & Hu, 2021a;b; Xu et al., 2022; Luo et al., 2022), among others. In contrast, flow-based generative methods (Liu et al., 2022; Albergo & Vanden-Eijnden, 2022; Lipman et al., 2022) present a more streamlined alternative to diffusion models (Sohl-Dickstein et al., 2015; Song & Ermon, 2019; Ho et al., 2020; Song et al., 2020), bypassing the need for forward and backward diffusion processes. Instead, they focus on noise-data interpolants (Albergo et al., 2023), simplifying the generative modeling process and potentially leading to more optimal probability paths with fewer sampling steps (Liu et al., 2022). Flow matching, which employs ODE-based continuous normalizing flows (Chen et al., 2018), further refines this approach. Conditional flow matching (CFM) (Lipman et al., 2022; Liu et al., 2022; Albergo & Vanden-Eijnden, 2022) learns the ODE that maps the probability path from the prior distribution to the target by regressing the push-forward vector field conditioned on individual data points. Riemannian flow matching (Chen & Lipman, 2023) extends CFM to operate on general manifolds, reducing the need for costly simulations (Ben-Hamu et al., 2022; De Bortoli et al., 2022; Huang et al., 2022). Recent advancements in the rectified flow framework (Liu et al., 2022), which focuses on learning linear interpolations between distributions, have demonstrated notable improvements in both efficiency (Liu et al., 2023) and quality (Esser et al., 2024; Yan et al., 2024), particularly in text-to-image generation. Our work aligns with these trends by exploring how to guide generation within this simple and general framework in a training-free manner (Ye et al., 2024).

**Multimodal generative models.** Diffusion models have demonstrated great success in modeling continuous data; however, many real-world applications involve multimodal data, such as tabular data (Kotelnikov et al., 2023), graph data (Qin et al., 2023a), and scientific data (Peng et al., 2023). A key challenge in this domain is generating discrete data within the diffusion framework. Unlike large language models (LLMs) (Achiam et al., 2023; Lin et al., 2024; Team et al., 2023; Ke et al., 2023), which excel at modeling discrete text, diffusion models have struggled in this area. Li et al. (2022) introduced continuous language diffusion models, which embed tokens in a latent space and use nearest-neighbor dequantization for generation. Subsequent research has enhanced performance through alternative loss functions (Han et al., 2022a; Mahabadi et al., 2023) and by incorporating conditional information, such as infilling masks (Gong et al., 2022; Dieleman et al., 2022). Gulrajani & Hashimoto (2024) further improved language diffusion models by making various refinements to the training process, allowing their performance to approach that of autoregressive LMs. While continuous approaches to discrete data modeling have seen advancements (Richemond et al., 2022; Han et al., 2022a; Chen et al., 2022; Strudel et al., 2022; Floto et al., 2023), discrete diffusion methods like D3PM (Austin et al., 2021) and subsequent works (Zheng et al., 2023; Chen et al., 2023; Ye et al., 2023) have demonstrated greater efficiency. Recent innovations, such as SEDD (Lou et al., 2023), extend score matching to discrete spaces, improving language modeling to a level competitive with autoregressive models. Additionally, DFM (Campbell et al., 2024) applies continuous-time Markov chains to enable discrete flow matching, contributing to the Multiflow framework by integrating continuous flow matching. Overall, multimodal generation is still an important open problem, and our work lays a theoretical foundation for multimodal guided flow, which will be beneficial for the study of multimodal generation.

**Generative Model for Molecular Generation.** Generative models have been applied to design drugs such as small molecules (Ramakrishnan et al., 2014; Axelrod & Gomez-Bombarelli, 2022; Francoeur et al., 2020; Irwin et al., 2020), proteins (Berman et al., 2000; Kryshtafovych et al., 2021; Haas et al., 2018), peptide (Wang et al., 2024), and antibodies (Jin et al., 2021). Our work focuses on designing small molecules, while the method can also be easily adapted to proteins, peptide and antibodies which requires multimodal generation. Molecular design can be categorized into target-agnostic design (Huang et al., 2023; Xu et al., 2022; 2023; Morehead & Cheng, 2024), where the goal is to generate valid sets of molecules without consideration for any biological target, and target-aware

design (Li et al., 2023; Masuda et al., 2020; Guan et al., 2023; Peng et al., 2022; Schneuing et al., 2022). Our work considers both types of molecule design tasks.

**Guidance for Diffusion and Flow Models.** Since the introduction of classifier guidance by Dhariwal & Nichol (2021), which employs a specialized time-dependent classifier for diffusion models, significant progress has been made in applying guidance to these models. Early research focused on straightforward objectives for linear inverse problems such as image super-resolution, deblurring, and inpainting(Chung & Ye, 2022; Wang et al., 2022; Zhu et al., 2023). These approaches were later extended by leveraging more flexible time-independent classifiers, achieved through approximations of the time-dependent score using methods like Tweedie's formula (Yu et al., 2022) and Monte Carlo sampling (Song et al., 2023). Recent methods, such as FreeDoM (Yu et al., 2023), UGD (Bansal et al., 2023), and TFG (Ye et al., 2024), incorporate various techniques to enhance the performance of training-free classifier guidance, leading to more advanced forms of guidance. However, some of these techniques lack clear theoretical support. A more recent work by Shen et al. (2024) aims to improve the performance of training-free guidance by drawing on ideas from adversarial robustness. Our work provides a rigorous theoretical framework for the design choices in training-free guidance, which we believe will serve as a strong foundation for future empirical research. In addition to classifier guidance, there are related works focusing on guiding the generation of specific diffusion models. For instance, DOODL (Wallace et al., 2023), DNO (Karunratanakul et al., 2024), and D-Flow (Ben-Hamu et al., 2024) utilize invertible models or flow models to backpropagate gradients to the latent noise. These methods emphasize noise optimization and often involve a training process tailored to a specific target, potentially making them slower than training-free classifier guidance. Additionally, RectifID (Sun et al., 2024) is a concurrent work exploring training-free guidance on rectified flow for personalized image generation. However, we have identified several theoretical issues in this work, which will be discussed in the following paragraph. This method addresses the training-free guidance via a fixed-point formulation, which is quite different from us and can only apply to continuous data.

## C.2 THEORETICAL ISSUES WITH RECTIFID.

Similar to many training-free guidance methods, RectifID aims to bypass the noise-aware classifier, which is typically trained according to the noise schedule of flow or diffusion models (as discussed in Equation 7 of their paper). They approach this by employing a fixed-point formulation:

$$z_1 = z_0 + v(z_1, 1) + s \cdot \nabla_{z_1} \log p(c|z_1), \tag{73}$$

which corresponds to Equation 8 in their paper. This equation is derived under the assumption that the rectified flow follows a linear path $z_t = z_0 + v(z_t, t)t$, which is strong. Since the velocity $v(z_t, t)$ is parameterized by a flow network and $\log p(c|z_1)$ can be estimated using a time-independent classifier, this formulation implies that $z_1$ can be obtained by solving a fixed-point problem. However, we think that when solving this fixed-point problem, the estimate of $\log p(c|z_1)$ from a time-independent classifier may be unreliable because some values of $z_1$ could be out-of-distribution for the classifier.

## C.3 THE MOTIVATION OF STUDYING TRAINING-FREE GUIDANCE

As training-free guidance is an emerging area in generative model research, this section highlights its significance as a critical and timely topic that warrants greater attention and effort, particularly in leveraging off-the-shelf models.

**Comparison with Conditional Generative Models.** Conditional generative models, such as Cond-EDM and Cond-Flow, require labeled data for training. This reliance on annotated datasets can limit training efficiency and scalability. In contrast, training-free guidance allows the use of any unconditional generative model, which can be trained in an unsupervised manner. This flexibility enables the scaling of generative models without the need for task-specific annotations. Furthermore, conditional generative models are tied to predefined tasks during training, making them unsuitable for new tasks post-training.

**Comparison with Classifier Guidance.** Classifier guidance (Dhariwal & Nichol, 2021) involves training a time-dependent classifier aligned with the noise schedule of the generative model. While this approach can utilize pre-trained unconditional flow or diffusion models, it still requires labeled data for classifier training on a per-task basis. This constraint makes it less adaptable for scenarios where the target is defined as a loss or energy function without associated data, or when leveraging pre-trained foundation models for guidance. In contrast, training-free guidance offers greater flexibility, allowing for broader and more dynamic applications without the need for task-specific classifier training.

**Comparison with Classifier-Free Guidance.** Classifier-free guidance (Ho & Salimans, 2022) has become a popular approach for building generative foundation models. However, it still requires task definitions to be determined prior to model training. A more versatile alternative is instruction tuning, where text instructions act as an interface for a variety of user-defined tasks. As shown by Ye et al. (2024), training-free guidance surpasses text-to-X models (e.g., text-to-image or text-to-video) in flexibility and robustness, as demonstrated by two failure cases of GPT-4. These models often struggle with interpreting complex targets in text form or constraining generated distributions using simple text prompts. Training-free guidance offers a more powerful and adaptable mechanism for controlling the generation process, enabling a broader range of user-defined targets and more precise control over outputs.

**Applications of training-free guidance.** In motif scaffolding, training-free guidance can direct the generated samples to possess a given motif without requiring a trained classifier (Didi et al., 2023). Similarly, in the *symmetric oligomer design* and *enzyme design with concave pockets* tasks in RFDiffusion (Watson et al., 2023)—a foundational model for protein design—training-free guidance is employed (referred to as "inference with external potentials"). Beyond biology, training-free guidance also extends to tasks in other domains, such as image, audio, and motion. Examples include deblurring, super-resolution, style transfer, audio declipping, obstacle avoidance, to name just a few. Notably, this approach eliminates the need for label-conditioned data collection or task-specific training when an off-the-shelf generator and target predictor are available. For further insights into its applications, we direct readers to TFG (Ye et al., 2024), which highlights 16 tasks across 40 targets leveraging training-free guidance.

## C.4 DISCUSSION OF CONTINUOUS PART MODELING IN MULTIFLOW

In this section, we examine the implicit assumptions underlying Multiflow (Campbell et al., 2024), which serves as a foundational basis for our method. While the derivation in Multiflow adopts a unified approach to modeling the continuous component $\boldsymbol{X}_{1|t}$ and the discrete component $\boldsymbol{a}_{1|t}$, its practical implementation differs significantly. Specifically, the likelihood of the discrete component $\boldsymbol{a}_{1|t}$ can be directly modeled using the output of a neural network, whereas the modeling of the continuous component $\boldsymbol{X}_{1|t}$ presents greater challenges.

Recall the key component of Multiflow is the computation of unconditional velocity Eq. (7). We copy it here for convenience.

$$\boldsymbol{v}_t(\boldsymbol{x}_t) = \mathbb{E}_{p_{1|t}(\boldsymbol{G}_1|\boldsymbol{G}_t)}\left[\boldsymbol{v}_{t|1}(\boldsymbol{x}_t|\boldsymbol{x}_1)\right]; \quad R_t(a_t, b) = \mathbb{E}_{p_{1|t}(\boldsymbol{G}_1|\boldsymbol{G}_t)}\left[R_{t|1}(a_t, b|a_1)\right].$$

To compute the continuous velocity $\boldsymbol{v}_t(\boldsymbol{x}_t)$, one could sample multiple $\boldsymbol{x}_{1|t}$ from $p_{1|t}(\cdot|\boldsymbol{G}_t)$ and estimate the expectation $\boldsymbol{v}_{t|1}(\boldsymbol{x}_t|\boldsymbol{x}_1)$ using Monte Carlo methods, as direct computation of the expectation involves intractable integrals. However, in Multiflow's and our implementation, the model predicts the expectation $\mathbb{E}_{1|t}[\boldsymbol{x}_{1|t}|\boldsymbol{G}_t]$. Consequently, $\boldsymbol{v}_t(\boldsymbol{x}_t) = \boldsymbol{v}_{t|1}(\boldsymbol{x}_t|\mathbb{E}_{1|t}[\boldsymbol{x}_{1|t}|\boldsymbol{G}_t])$, where $\mathbb{E}_{1|t}[\boldsymbol{x}_{1|t}|\boldsymbol{G}_t]$ is parameterized by the output of the flow model (a neural network).

This assumption does not compromise the theoretical validity of our theorems or those in the Multiflow paper since $\boldsymbol{v}_{t|1}$ is linear *w.r.t.* $\boldsymbol{x}_1$ as defined in Eq. (6): $\boldsymbol{v}_{t|1}(\boldsymbol{x}_t|\boldsymbol{x}_1) = \frac{\boldsymbol{x}_1-\boldsymbol{x}_t}{1-t}$. And computing the expectation of $\mathbb{E}_{p_{1|t}(\boldsymbol{G}_1|\boldsymbol{G}_t)}\boldsymbol{v}_{t|1}(\boldsymbol{x}_t|\boldsymbol{x}_1)$ is equivalent to $\boldsymbol{v}_{t|1}(\boldsymbol{x}_t|\mathbb{E}_{1|t}[\boldsymbol{x}_{1|t}|\boldsymbol{G}_t])$. Therefore, the linearity allows us to model the expectation only.

# D EXPERIMENTAL DETAILS

## D.1 TARGET-AGNOSTIC DESIGN SETTING

We conduct experiments in this setting following EDM (Hoogeboom et al., 2022), where the generative process is by first sampling $c, n \sim p(c, n)$, where $c$ is the target property and $n$ is the number of atoms, and then samples the molecule $G$ given these two information. We compute $c, n \sim p(c, n)$ on the training partition as a parameterized two dimensional categorical distribution where we discretize the continuous variable $c$ into small uniformly distributed intervals.

For structure similarity, we use all the structures in the unseen half of training set for generative model and guidance predictor. The Tanimoto coefficient results are averaged over all the structures.

## D.2 IMPLEMENTATION OF TFG-FLOW

**Network configurations.** To train Multiflow on target-agnostic small molecular generation, we follow EDM (Hoogeboom et al., 2022) to use an EGNN with 9 layers, 256 features per hidden layer and SiLU activation functions. We use the Adam optimizer with learning rate $10^{-4}$ and batch size 256. Only atom types (discrete) and coordinates (continuous) have been modelled, which is different from unconditional EDM that includes atom charges. All models have been trained for 1200 epochs (for QM9) and 20 epochs (for GEOM-Drug and CrossDocked) while doing early stopping by evaluating the validation loss. For target-aware molecular design, we follow TargetDiff (Guan et al., 2023) to use EGNN with $k$NN graph, where $k = 32$ and reduces the batch size to 16 and the hidden layer dimension to 128. For target predictors, we train a 6-layer discriminative EGNN by adding linear head on the average pooling of output node feature. In our implementation, we only model heavy atoms.

**Hyperparameter searching.** As noted by TFG (Ye et al., 2024), hyperparameter searching is the key problem for training-free guidance. TFG designs complicated hyperparameter space where three hyperparameters (which consist of a scalar and a scheduling strategy) and two iteration parameters should be determined. Our TFG-Flow only has the four hyperparameters $N_{\text{iter}}, \rho, \tau, K$ to be tuned, and we show that a logarithmic scale of $K$ is sufficient, and $N_{\text{iter}}$ can be set according to the computational resources. So we fix $N_{\text{iter}} = 4$ and $K = 512$ in our experiments, while grid search the $\rho$ and $\tau$ for different applications. The search space is defined by first find the appropriate magnitude, as different applications have varied order of value magnitude. After determining the appropriate magnitude, we double $\rho$ and $\tau$ for 4 times to construct the search space. For example, the appropriate search space for polarizability is $(\rho, \tau)$ is $\{0.01, 0.02, 0.04, 0.08\} \times \{10, 20, 40, 80\}$. We then search the space via small scale generation (the number of 512 following Ye et al. (2024)). The search target is to minimize the guidance performance (e.g., MAE in quantum property guidance), while keeping a validity over 75%.

## D.3 BASELINES

We introduce the baselines used in this paper here. For baselines in target-agnostic molecular design,

- **Upper bound**: This baseline is from EDM (Hoogeboom et al., 2022), which removes any relation between molecule and property by shuffling the property labels in the unseen half of QM9 training set for oracle target predictor, and evaluate MAE on it. If a baseline outperforms upper bound, then it should be able to incorporate conditional property information into the generated molecules.

- **#Atoms**: "#Atoms" predicts the molecular properties by only using the number of atoms in the molecule. If a baseline outperforms "#Atoms", it should be able to incorporate conditional property information into the generated molecular structure beyond the number of atoms.

- **Lower bound**: The MAE of directly predicting property using the oracle target predictor.

- **Cond-EDM**: The conditional EDM is implemented by simply inputting a property $c$ to the neural network of EDM (Hoogeboom et al., 2022), without changing the diffusion loss.

- **EEGSDE**: EEGSDE (Bao et al., 2022) trains a time-dependent classifier to guide the conditional EDM. This is the best method for guided generation in quantum guidance task.

- **DPS**: DPS (Chung & Ye, 2022) estimate the time-dependent gradient $\nabla_{\boldsymbol{x}_t} \log p(\boldsymbol{x}_t, t)$ with $\nabla_{\boldsymbol{x}_t} \log p(\boldsymbol{x}_{0|t})$ using Tweedie's formula (here $\boldsymbol{x}_0$ means the clean data in diffusion formulation, which is different from flow where $\boldsymbol{x}_1$ is the clean data).

- **LGD**: LGD (Song et al., 2023) replaces the point estimation $\boldsymbol{x}_{0|t}$ with Gaussian kernel $\mathcal{N}(\boldsymbol{x}_{0|t}, \sigma^2 \boldsymbol{I})$. LGD uses Monte-Carlo sampling to estimate the expectation of $\nabla_{\boldsymbol{x}_t} \log p(\boldsymbol{x}_{0|t})$.

- **FreeDoM**: FreeDoM (Yu et al., 2023) generalizes DPS by introducing a "time-travel strategy" that iteratively denoises $\boldsymbol{x}_{t-1}$ from $\boldsymbol{x}_t$ and adds noise to $\boldsymbol{x}_{t-1}$ to regenerate $\boldsymbol{x}_t$ back and forth.

- **MPGD**: MPGD (He et al., 2024) computes the gradient $\nabla_{\boldsymbol{x}_{0|t}} \log f(\boldsymbol{x}_{0|t})$ instead of $\nabla_{\boldsymbol{x}_t} \log f(\boldsymbol{x}_{0|t})$ to avoid back-propagation through the diffusion model.

- **UGD**: UGD (Bansal et al., 2023) is similar to FreeDoM, which additionally solves a backward optimization problem and guides $\boldsymbol{x}_{0|t}$ and $\boldsymbol{x}_t$ simultaneously.

- **TFG**: TFG (Ye et al., 2024) unifies all the techniques in training-free guidance, and turn the problem into hyperparameter searching.

- **Cond-Flow**: Conditional flow is implemented by inputting the target value of property $c$ to Multiflow. The modification is the same as Cond-EDM with respect to EDM.

For baselines in target-aware molecular design,

- **AR**: AR (Luo et al., 2021) is based on GNN and generates atoms into a protein pocket in an autoregressive manner.

- **Pocket2Mol**: Pocket2Mol (Peng et al., 2022) generates atoms sequentially but in a more fine-grained manner by predicting and sampling atoms from frontiers.

- **TargetDiff**: TargetDiff (Guan et al., 2023) generates the molecules via diffusion, which is similar to EDM that incorporates pocket information as inputs.

- **Multiflow**: We follow TargetDiff to adapt target-agnostic Multiflow to target-aware Multiflow. The simplex $\Gamma$ projection becomes to substracting the center of gravity $w.r.t.$ the protein pocket.

### D.4 DATASET PREPROCESSING

We introduce the dataset information for our expeimernts.

**QM9.** The QM9 dataset is a widely-used resource for benchmarking models in the field of molecular generation, particularly for 3D structures. This dataset comprises about 134,000 molecules, each containing up to 9 heavy atoms (not counting hydrogen atoms), derived from a subset of the GDB-17 database which itself is a database of 166 billion synthetic molecules. Each molecule in the QM9 dataset is characterized by its chemical properties and 3D coordinates of atoms. In the specific usage of QM9 as mentioned, the dataset was divided into training, validation, and test sets with sizes of 100,000, 18,000, and 13,000 molecules, respectively, following the setup used by the EDM. We use the QM9 dataset from Huggingface Hub[6].

**GEOM-Drug.** The GEOM-Drug dataset is tailored for applications in drug discovery, providing a more complex and realistic set of molecules for the development and benchmarking of molecular generation models. This dataset was specifically chosen to train and evaluate models that are intended to simulate and understand real-world scenarios in drug design. GEOM-Drug differs from simpler datasets like QM9 by focusing on molecules that are more representative of actual drug compounds. These molecules often contain major elements found in drugs, such as carbon (C), nitrogen (N), oxygen (O), fluorine (F), phosphorus (P), sulfur (S), and chlorine (Cl). We follow MolDiff (Peng et al., 2023) to exclude hydrogen atoms and minor element types such as boron (B), bromine (Br),

---

[6]https://huggingface.co/datasets/yairschiff/qm9

iodine (I), silicon (Si), and bismuth (Bi). We use the opensourced version of GEOM-Drug processed by MolDiff codebase[7], which filters the molecules through several criterion. The selection criteria were as follows:

- molecules must be compatible with the RDKit package;
- structures should be intact;
- molecules should contain between 8 and 60 heavy atoms;
- only elements such as H, C, N, O, F, P, S, and Cl were allowed;
- permissible chemical bonds included single, double, triple, and aromatic bonds.

Following these criteria, hydrogen atoms were excluded, and the molecules were divided into training, validation, and testing datasets containing 231,523, 28,941, and 28,940 molecules, respectively.

**CrossDocked2020.** The Crossdocked dataset initially comprises 22.5 million poses of small molecules docked to proteins, featuring 2,922 unique protein pockets and 13,839 unique small molecules. We utilized non-redundant subsets that were filtered and processed from the TargetDiff codebase[8]. These subsets include 100,000 protein-molecule pairs for training and 100 pairs for testing.

## D.5 EVALUATION METRICS

We introduce the evaluation metrics used in this paper.

- **MAE (Mean Absolute Error)**: A measure of the average magnitude of the absolute errors between the predicted values and the actual values, without considering their direction. It's calculated as the average of the absolute differences between the predictions and the actual outcomes, providing a straightforward metric for regression model accuracy.
- **Tanimoto coefficient**: A metric used to compare the similarity and diversity of sample sets. It is defined as the ratio of the intersection of two sets to their union, often used in the context of comparing chemical fingerprints in computational chemistry.
- **Vina score**: A scoring function specifically designed for predicting the docking of molecules, particularly useful in drug discovery. It estimates the binding affinity between two molecules, such as a drug and a receptor, helping in the identification of potential therapeutics.
- **SA score (Synthetic Accessibility score)**: A heuristic measure to estimate the ease of synthesis of a given molecular structure. Lower scores indicate molecules that are easier to synthesize, useful in the design of new compounds in medicinal chemistry.
- **QED (Quantitative Estimate of Drug-likeness)**: A metric that quantifies the drug-likeness of a molecular structure based on its physicochemical properties and structural features, aiming to identify compounds that have desirable attributes of a potential drug.

We also test other metrics related to generation quality following EDM (Hoogeboom et al., 2022) and MolDiff (Peng et al., 2023) in App. E:

- **Validity**: Measures the percentage of generated molecules that are chemically valid (can be parsed by RDKit).
- **Atom stability**: Assesses whether the atoms have the right valency.
- **Molecular stability**: Assesses whether all the atoms are stable in the molecule.
- **Uniqueness**: Quantifies the uniqueness of generated molecules.
- **Novelty**: Measures how novel the generated molecules are when compared to the training set.
- **Connectivity**: Checks if all atoms are connected in the molecule. We use the lookup table from EDM to infer the chemical bonds.

---

[7] https://github.com/pengxingang/MolDiff
[8] https://github.com/guanjq/targetdiff

### D.6 LEVERAGING PRE-TRAINED FOUNDATION MODELS AS TARGET PREDICTOR

In our main experiments, the target predictors were trained specifically for guidance purposes to ensure a direct and fair comparison with existing baselines such as TFG (Ye et al., 2024). However, a significant advantage of training-free guidance methods like **TFG-Flow** is their ability to leverage *pre-trained foundation models* as target predictors without additional training. This capability not only enhances performance but also reduces computational resources and time.

To demonstrate this advantage, we conducted additional experiments using **UniMol** (Zhou et al., 2023), an open-source universal 3D molecular foundation model applicable to various downstream tasks. UniMol is a powerful pre-trained model that can predict a wide range of molecular properties. We used the fine-tuned UniMol model to predict molecular properties, serving as our guidance network in the experiments corresponding to Table 5.

Table 5: Conditional generation results on the QM9 dataset using pre-trained UniMol as the target predictor. Lower values are better for all metrics.

| Method | $C_v \downarrow$ | $\mu \downarrow$ | $\alpha \downarrow$ | $\Delta\epsilon \downarrow$ | $\epsilon_{\text{HOMO}} \downarrow$ | $\epsilon_{\text{LUMO}} \downarrow$ |
|---|---|---|---|---|---|---|
| Cond-Flow | 1.52 | 0.962 | 3.10 | 805 | 435 | 693 |
| TFG-Flow (Vanilla predictor) | 1.48 | 0.880 | 3.52 | 917 | 364 | 998 |
| TFG-Flow (UniMol) | 1.12 | 0.802 | 2.89 | 585 | 312 | 587 |
| EEGSDE | 0.941 | 0.777 | 2.50 | 487 | 302 | 447 |

As shown in Table 5, incorporating a strong pre-trained target predictor like UniMol significantly enhances the performance of TFG-Flow across all evaluated molecular properties. Specifically, the Mean Absolute Error (MAE) decreases notably when using UniMol compared to the vanilla predictor trained specifically for guidance. The performance gap between our training-free TFG-Flow (with UniMol) and the training-based EEGSDE, which requires training both a conditional generative model and a time-dependent predictor, has notably decreased.

These findings highlight several key advantages:

- **Improved Performance.** The integration of a strong pre-trained model like UniMol boosts the performance of TFG-Flow, achieving results closer to or competitive with training-based methods.
- **Flexibility.** Training-free guidance methods can seamlessly incorporate any off-the-shelf pre-trained models, including those not originally designed for the guidance task. This flexibility is particularly beneficial when high-quality pre-trained models are available.
- **Reduced Computational Effort.** By leveraging pre-trained models, we eliminate the need to train additional target predictors or time-dependent classifiers, saving both time and computational resources.

Our experiments demonstrate that TFG-Flow benefits significantly from incorporating pre-trained foundation models as target predictors. This capability enhances performance, narrows the gap with training-based methods, and underscores the practical advantages of training-free guidance.

### D.7 HARDWARE AND SOFTWARE SPECIFICATIONS

We run most of the experiments on clusters using NVIDIA A800s with 128 CPU cores and 1T RAM. We implemented our experiments using PyTorch, RDKit, and the HuggingFace library. Our operating system is based on Ubuntu 20.04 LTS.

# E   DETAILED EXPERIMENTAL RESULTS

## E.1   CONDITIONAL FLOW

Table 6: The full results for conditional flow (Cond-flow) on QM9 quantum property guidance. The results are averaged over 3 random seeds, where standard deviations are reported in Table 7.

|  | Validity↑ | Uniqueness↑ | Novelty↑ | Mol. stability↑ | Atom stability↑ | Connectivity↑ | MAE↓ |
|---|---|---|---|---|---|---|---|
| $\alpha$ | 78.03% | 98.78% | 87.75% | 89.54% | 98.49% | 69.55% | 3.10 |
| $\mu$ | 83.60% | 97.81% | 86.71% | 92.87% | 99.12% | 74.40% | 0.96 |
| $C_v$ | 64.96% | 98.78% | 85.48% | 81.20% | 96.89% | 57.91% | 1.52 |
| $\Delta\varepsilon$ | 75.95% | 97.42% | 87.26% | 88.41% | 98.52% | 64.86% | 804.03 |
| $\varepsilon_{\text{HOMO}}$ | 87.93% | 95.90% | 89.14% | 94.51% | 99.32% | 73.72% | 436.02 |
| $\varepsilon_{\text{LUMO}}$ | 80.76% | 98.46% | 90.57% | 91.58% | 99.00% | 67.42% | 692.78 |

Table 7: The standard deviations for conditional flow (Cond-flow) on QM9 quantum property guidance.

|  | Validity | Uniqueness | Novelty | Mol. stability | Atom stability | Connectivity | MAE |
|---|---|---|---|---|---|---|---|
| $\alpha$ | 0.5921 | 0.2458 | 0.4712 | 0.4000 | 0.0551 | 0.6409 | 0.0231 |
| $\mu$ | 0.7703 | 0.1607 | 0.4443 | 0.6853 | 0.0808 | 0.4022 | 0.0034 |
| $C_v$ | 0.6730 | 0.2427 | 0.5601 | 1.0701 | 0.2290 | 0.5462 | 0.0467 |
| $\Delta\varepsilon$ | 0.4349 | 0.2875 | 0.3831 | 0.2139 | 0.0404 | 0.6322 | 12.74 |
| $\varepsilon_{\text{HOMO}}$ | 0.1553 | 0.4576 | 0.7246 | 0.2464 | 0.0252 | 0.6218 | 9.054 |
| $\varepsilon_{\text{LUMO}}$ | 0.4306 | 0.1234 | 0.5103 | 0.3798 | 0.0351 | 0.2207 | 9.139 |

## E.2 TFG-FLOW

Table 8: The full results for TFG-Flow on QM9 quantum property guidance. The results are averaged over 3 random seeds, where standard deviations are reported in Table 9.

|  | Validity↑ | Uniqueness↑ | Novelty↑ | Mol. stability↑ | Atom stability↑ | Connectivity↑ | MAE↓ |
|---|---|---|---|---|---|---|---|
| $\alpha$ | 76.31% | 99.43% | 93.44% | 89.89% | 98.75% | 66.40% | 3.52 |
| $\mu$ | 75.34% | 99.01% | 89.89% | 86.85% | 98.33% | 63.64% | 0.88 |
| $C_v$ | 75.56% | 98.99% | 90.89% | 87.78% | 98.43% | 68.55% | 1.48 |
| $\Delta\varepsilon$ | 75.95% | 97.42% | 87.26% | 88.41% | 98.52% | 64.86% | 914 |
| $\varepsilon_{\mathrm{HOMO}}$ | 87.93% | 95.90% | 89.14% | 94.51% | 99.32% | 73.72% | 364 |
| $\varepsilon_{\mathrm{LUMO}}$ | 80.76% | 98.46% | 90.57% | 91.58% | 99.00% | 67.42% | 998 |

Table 9: The standard deviations for TFG-Flow on QM9 quantum property guidance.

|  | Validity | Uniqueness | Novelty | Mol. stability | Atom stability | Connectivity | MAE |
|---|---|---|---|---|---|---|---|
| $\alpha$ | 0.8051 | 0.1015 | 0.4419 | 0.7518 | 0.0900 | 1.0564 | 0.0523 |
| $\mu$ | 0.0693 | 0.1361 | 0.4844 | 0.3800 | 0.0751 | 0.3166 | 0.0164 |
| $C_v$ | 0.6660 | 0.2127 | 0.5501 | 1.0601 | 0.2390 | 0.3462 | 0.0455 |
| $\Delta\varepsilon$ | 0.6673 | 0.1058 | 0.2498 | 0.4743 | 0.3528 | 0.3407 | 20.7743 |
| $\varepsilon_{\mathrm{HOMO}}$ | 0.4603 | 0.1501 | 0.2425 | 0.1997 | 0.0416 | 0.2761 | 7.4903 |
| $\varepsilon_{\mathrm{LUMO}}$ | 0.1000 | 0.1501 | 0.1102 | 0.2589 | 0.0173 | 0.2454 | 14.9410 |

