# OpenReview forum: "TFG-Flow: Training-free Guidance in Multimodal Generative Flow"
_ICLR.cc/2025/Conference — ICLR 2025 Poster_

### Official Review · Reviewer_Pmbj · 2024-10-27

**Soundness:** 3
**Presentation:** 2
**Contribution:** 3
**Rating:** 8
**Confidence:** 2

**Summary:**

In this paper, the authors present a guidance approach multi-modal continuous normalizing flows trained using flow matching. This field is currently very active, but the authors present a new formal approach to this problem, which they claim does not need any further training: Using a "foundation model," (modeling p(x)) and an off-the-shelf predictive model (modeling y=f(x)), they generate conditional samples p(x|y).
Minor comment:
The transport/continuity equation is repeatedly referred to as the Fokker-Planck equation.

**Strengths:**

- Strong formal framework. Broad applicability beyond molecular examples shown in this work.
	- Good results, and transparency about limitations and cases where performance is less good.
	- Several experiments targeting different molecular problems.

**Weaknesses:**

- Main condition for the paper is training free guidance, all the benchmarks shown where the results stand out, use models which are trained for the purpose guidance.
        - Presentation suffers somewhat from being overly compact.

**Questions:**

- One of the key contributions is Theorem 3.4, which provides a sampling estimator for the conditional rate matrix (guided discrete flow). The scaling property is good compared to the baseline, but may still be prohibitive in many applications, e.g. experimental design. How does, the variance of the estimator grow if k<K samples are used? Are there some ways to further improve on this estimator to make it useful in other application domains?

---

### Official Review · Reviewer_U4d6 · 2024-10-31

**Soundness:** 3
**Presentation:** 3
**Contribution:** 2
**Rating:** 5
**Confidence:** 3

**Summary:**

The authors tackle the problem of training-free guidance in multi-modal generative flow. Specifically, the work builds on the recent development of "Multiflow", i.e. a flow-based generative model for mixed modality data (i.e. discrete and continuous). This paper describes how one can perform guidance in such models without any additional training and only requiring access to time-independent property predictor. Emprically, superiority over other training-free guidance methods is demonstrated.

**Strengths:**

* I think that the paper is very well written. The problem setup and motivation are clearly explained; the method and contributions are described in a way that is easy to understand.
* Theoretical contributions, which are precisely stated and proven.
* Encouraging experimental results.
* Submitted code for improved reproducibility.

**Weaknesses:**

* To me, the biggest weakness lies in the following missing piece of the explanation of the "mutlimodal flow model". My understanding of the base model is the following. We train the model $p_{1|t}(G_1 | G_t)$, we then use it to estimate marginal velocities described in Equations (7), (9) and (10). This allows us to take a small step to estimate $G_{t + dt}$ based on $G_t$. This implicitly implies that $p_{1|t}$ is efficient to sample from. Otherwise, this procedure would be very expensive. My question is the following: "If we know how to efficiently sample from $p_{1|t}$, why bother with the flow model at all? Why not just sample $G_0 \sim p_0$ and directly sample $G_1 \sim p_{1|0}(G_1 | G_0)$ ? I read the paper that authors build on [1], but I do not think this is explained properly there either. Here are follow-up questions:
  * How is $p_{1|t}$ parametrized? I do not mean what architecture of the model, but what kind of a model? Is this a generative model? If so, what kind? (GAN, NF, VAE, sth else?).
  * The marginal flow is given by an expectation under $p_{1|t}$. I assume that this is approximated using Monte Carlo estimation? How many samples are used? What variance does this exhibit?
  * In lines 196-197 you say that $g$ parametrizes this model and for input $G_t$ returns "sample/likelihood of $G_1$". What does the slash "/" mean? Does it return both the sample and the corresponding likelihood? I think this narrows down the choices for $p_{1|t}$, because not all generative models can return do both: generate samples and their corresponding likelihoods.
  * If one can efficiently sample from $p_{1|t}$ why not just generate a collection of $G_1 \sim p_{1|t}(G_1|G_t)$ and choose the one with the highest value of our property predictor $f$? Have the authors considered such a baseline? How would this compare to the proposed method?  think that this is very closely related to what authors are actually doing implicitly (see Question 2 is the questions section below)
* Benchmark model size/runtime comparison in Tables 1, 2 and 4. I think that the model comparison would be much more meaningful if we had the following information:
  * What are the model sizes?
  * What is the runtime of each model, both training and inference?
* Lack of comparison with optimization-based structure-based drug design methods. In table 4, I would strongly suggest comparing with optimization-based method like Autogrow4. E.g. in [3] it was shown to be a very strong baseline and it makes a fair comparison, because the method you are proposing is an optimization-based method and not merely generative one like the ones used as baselines.

---

References

[1] Campbell et al "Generative Flows on Discrete State-Spaces: Enabling Multimodal Flows with Applications to Protein Co-Design" (ICML 2024)

[2] Spiegel et al. "Autogrow4: an open-source genetic algorithm for de novo drug design and lead optimization" (ChemInf 2020)

[3] Karczewski et al. "WHAT AILS GENERATIVE STRUCTURE-BASED DRUG DESIGN: TOO LITTLE OR TOO MUCH EXPRESSIVITY?" (Arxiv)

**Questions:**

* Line 317: "Mathematically, we expect...". Why do we expect this? Could the authors elaborate? Is this a heuristic with desirable properties (favouring larger values of $f$) or is this more principled?
* Line 319: "One can show..." I am not sure that this is true. To me this looks more like gradient ascent of $x \to \log f (\mathbb{E}(X_1|X_t=x, a_t))$ than generating samples according to some distribution. Could the authors elaborate?
* Table 1: I think that highlighting the results for TFG-Flow in bold is misleading. I was under the impression that this is the best method overall and not only among training-free methods. I think that the authors should make the rank of methods more transparent. This also goes back to the point of Weakness (2): Perhaps the training-free methods are not even faster, since we do not know the runtime of the baselines and the proposed method, I do not think this is justified to evaluate them separately (i.e. TFG-Flow should not be in bold since it is significantly worse than EEGSDE). If, however, it is much faster due to being training-free than one can argue why they might be considered in different categories.
* Lines 470, 473:  The authors say that they train the drug quality prediction network, but then they say that there is no need to train the oracle target predictor. So what is trained and what does not need to be trained?
* Line 479 - Is this a reasonable definition of the target? Do we want to match the quality to the reference or do we want to maximize it? The current formulation looks like it wants to match (maximized at $\mathcal{E}(G) = c$). Why not maximize?
* Line 1499 - why is this assumption incorrect?

---

> ### Comment · Reviewer_U4d6 · 2024-11-21
> **Reply 1/2**
>
> Thank you for your detailed response. I address the points below.
>
> * *In practice, people typically find that taking smaller steps leads to higher quality samples [a,b]*
>
> I disagree that referencing flow matching here is relevant. In the flow matching model taking smaller steps is justified, because the flow-matching model is a continuous normalizing flow (CNF), i.e. sampling is given by solving an ODE. Therefore it is well understood how taking smaller steps in solving an ODE leads to a smaller error. There is no $p_{1|t}$ in flow matching.
>
> * *$p_{1|t}$ is a sub-moudule implemented as a neural network. It takes both the continous and discrete parts of as input, and outputs a sample $X_t$ for the continous part*
>
> I am afraid this does not answer my question. I asked how the **distribution** $p_{1|t}$ is defined. "Submodule", nor "neural network" does not define a distribution. The discrete part is clear, you output a probability distribution over an entire discrete space. How about the continuous part? The notations and derivations assume that $p_{1|t}$ is a distribution. How is this distribution defined? Saying that "it is a submodule implemented as a neural network that outputs a sample" does not define a distribution. Where does the randomness come from if it is a distribution? For example, if the neural also took noise as input, then it would define an implicit distribution (like GANs). However, Eq (8) in [1] suggests that $p_{1|t}$ is trained with maximum likelihood. That rules out GANs. I presume that the authors use $p_{1|t}(x_1|x_t) = \mathcal{N}(x_1 | \mu(x_t), \sigma(x_t)I)$, where $\mu$ is the neural network that the authors refer to. However, I do not see this defined anywhere in your paper, this reponse, nor in [1]. An example of how one might define a distribution is e.g. the definition of $p_{t|1}$ (reverse conditioning) in Eq (11) in [1]. Not every neural network that takes some input and produces some output defines a probability distribution (unless the authors mean a Dirac delta distribution, but I would need to check if the derivations hold in that case; although this is also not stated in the text)
>
> * *In [1], the authors use a single sample for the Monte Carlo estimation*
>
> In your paper do you also use a single sample? I also asked about the variance.
>
> *Your suggestion seems to align with a “generate-then-filter” approach, where multiple samples are generated, and the top-1 sample with the highest target value is selected*
>
> This is not what I am suggesting. I do not mean sampling $G_1 \sim p_1$. I mean in TGF-Flow, instead of generating any random sample $G_1 \sim p_{1|t}$, sampling multiple $G_1 \sim p_{1|t}$ and choosing the best one before moving $t \to t + dt$.
>
> *the merit of training-free guidance is not “efficiency” in terms of speed or runtime.*
>
> If runtime is not the issue, why not train additional classifiers and use them for guidance? In that case, I am not sure I understand the motivation of the study. What is the added value of the method being "training free" if it's not efficiency?
>
> *Table 1 runtime and parameter count*
>
> Do I understand correctly that all the baselines and the proposed method share the exact same architecture? Regarding training times, what does it mean "e.g., 1h10min for training the target predictor and 2h for training the generative models)". Does every method require training a target predictor? Let me clarify what I meant. It would be helpful if you provided a table that summarizes the total training time, total parameter count, and total sampling time for each method, so that every component of every model is included in all three categories. E.g. is none of the baseline methods using regular (classifier-free or classifier-based) guidance? If not, why not? This would be a reasonable baseline. If yes, then I would be surprised if they require exactly the same training time as those that do not require this.

---

> ### Comment · Reviewer_U4d6 · 2024-11-21
> **Reply 2/2**
>
> * *The performance of training-based guidance methods (e.g., EEGSDE) typically serves as an upper bound for training-free methods [...] the merit of training-free methods lies not in their efficiency or speed but in their flexibility and independence from time-dependent classifiers*
>
> This is what I think I'm missing. To reiterate the point above: If speed is not a concern then I don't see the gain in pursuing training-free guidance methods. I can just spend X amount of additional time to train the time-dependent classifier and use regular guidance. Especially since the performance gap to the training-based remains significant.
>
> I would also reiterate the point I made in the original review. I would suggest changing the presentation in Table 2. In the current revision it still implies that the proposed method is the best overall. It is not the case. It is significantly worse than EEGSDE. The only gain is that TGF-Flow is "training-free" (but not necessarily faster).
>
> *To demonstrate the benefits of the training-free property to leverage flexible target predictors, we included results using an open-sourced foundation model UniMol [2] as the target predictor for experiments in Table 1. UniMol, a strong off-the-shelf model, significantly enhances the performance of TFG-Flow, narrowing the gap with EEGSDE*
>
> But the performance is still worse than EEGSDE. Respectfully, I do not see how this demonstrates the benefits of the training-free property. Just to clarify, does EEGSDE also use UniMol as guidance in this experiment? (What I mean is an additional, perhaps time-dependent, classifier trained to guide EEGSDE towards the signal from UniMol?)
>
> * *we incorporated a comparison with optimization-based methods*
>
> Could you please clarify two things for me? In this table, is "time" measuring the time to generate a single molecule? I am curious about the runtime, because you mention in the generate-then-filter you first generate 10 candidate molecules and then choose the best one. In [2] it is reported that estimating Vina score for a single protein-ligand complex takes 20 seconds, so this method should at least take 200 seconds to produce a single molecule. Does Vina2 run faster on your machine?
>
> * *The claim on the distributional property follows from existing works [4,5,6].*
>
> Could the authors point me to the specific derivation/equation/claim in e.g. [4] where I can find that derivation?
>
> * *To clarify, we train the drug quality prediction network to guide the flow model during training*
>
> I am afraid this confused me more. So what does training-free mean? Since the guidance is needed during training?
>
> * *The subtlety here is that $z_t$ is a function of $t$ .*
>
> Again, I must respectfully disagree. I went over the derivation in [3] and I do not think what you are referring to is an error, but rather a slight abuse of notation. Oftentimes, when one writes $\nabla_{z_t}f(z_t, t)$ it actually means $\nabla_z f(z, t)|_{z=z_t}$, which is clearly what the authors meant given the preceding derivation. In that case $z$ and $t$ are independent variables and $t$ is treated as constant when taking the gradient w.r.t. $z$.
>
> However, funnily enough, I do believe that [3] makes a critical error in their reasoning. It is however in Equation (3), not in Equation (9). The authors of [3] assume that a "well-trained" flow is linear. This is clearly not true. I will reach out to the authors of [3] for an explanation.
>
> * *Table 5*
>
> Please fix the citations in Table 5. Neither RGA, nor 3D-MCTS were developed by [2].
>
> ---
>
> [1] Campbell et al. "Generative Flows on Discrete State-Spaces: Enabling Multimodal Flows with Applications to Protein Co-Design" (ICML 2024)
>
> [2] Karczewski et al. "What Ails Generative Structure-Based Drug Design Too Little or Too Much Expressivity?" (Arxiv)
>
> [3] Sun et al. "RectifID: Personalizing Rectified Flow with Anchored Classifier Guidance" (NeurIPS 2024)

---

### Official Review · Reviewer_oDU6 · 2024-10-31

**Soundness:** 2
**Presentation:** 2
**Contribution:** 2
**Rating:** 6
**Confidence:** 5

**Summary:**

This paper studies the conditional generation problem in the context of molecular generation. Specifically, the authors propose a "classifier-guidance" approach for inherently multimodal molecular structures (atom token and coordinate) with flow matching/rectified flow-based generative models. Experiments are conducted on several molecular optimization problems.

**Strengths:**

* Conditional generation is an important and practical problem in molecular design. In fact, previous work in this space (recent trend of diffusion-based generative models) has under-studied how to do conditional generation and molecular optimization.
* The proposed method is straightforward and effective based on validations from experiments.

**Weaknesses:**

* Even though conditional generation is under-studied in the small molecular generation domain, it is well-studied in other domains (especially on exact guidance; see questions). [1,2,3]
* Molecular optimization has been a long-standing challenge in this field and unfortunately, almost none of them are mentioned or compared, see the review [4] for references.

[1] Lu, C., Chen, H., Chen, J., Su, H., Li, C. and Zhu, J., 2023, July. Contrastive energy prediction for exact energy-guided diffusion sampling in offline reinforcement learning. In International Conference on Machine Learning (pp. 22825-22855). PMLR.

[2] Wu, L., Trippe, B., Naesseth, C., Blei, D. and Cunningham, J.P., 2024. Practical and asymptotically exact conditional sampling in diffusion models. Advances in Neural Information Processing Systems, 36.

[3] Zhao, S., Brekelmans, R., Makhzani, A. and Grosse, R.B., Probabilistic Inference in Language Models via Twisted Sequential Monte Carlo. In Forty-first International Conference on Machine Learning.

[4] Du, Y., Jamasb, A.R., Guo, J., Fu, T., Harris, C., Wang, Y., Duan, C., Liò, P., Schwaller, P. and Blundell, T.L., 2024. Machine learning-aided generative molecular design. Nature Machine Intelligence, pp.1-16.

**Questions:**

* Is the guidance exact? i.e. if you write down the conditional vector field of the target distribution, is it equal to Eq 15? It seems you only have the term for guidance without the original vector field. In addition, in the diffusion literature, it is known that evaluating the guidance gradient as $\nabla E(\mathbb{E}_{X_1|X_t}(X_1))$ is not exact, where $E$ is the energy here, e.g. see [1].
* for Eq 15, should it be $X_{t+1} \leftarrow X_{t} + \cdots$?
* The rate matrix R is not properly defined, i.e. what does $R(\cdot, \cdot)$ mean?
* In Sec 3.2, why is the expectation calculation exponentially growing? Since you only need to model for each atom their rate matrix, is it in the order of $n|A|$?
* Is the Monte Carlo estimator for Eq 12 biased?
* In structured-based drug design, there exist methods for property optimization, e.g. see [2] using an evolutionary algorithm.

[1] Lu, C., Chen, H., Chen, J., Su, H., Li, C. and Zhu, J., 2023, July. Contrastive energy prediction for exact energy-guided diffusion sampling in offline reinforcement learning. In International Conference on Machine Learning (pp. 22825-22855). PMLR.

[2] Schneuing, A., Harris, C., Du, Y., Didi, K., Jamasb, A., Igashov, I., Du, W., Gomes, C., Blundell, T., Lio, P. and Welling, M., 2022. Structure-based drug design with equivariant diffusion models. arXiv preprint arXiv:2210.13695.

---

### Official Review · Reviewer_t7Ci · 2024-11-03

**Soundness:** 2
**Presentation:** 2
**Contribution:** 2
**Rating:** 6
**Confidence:** 2

**Summary:**

The authors develop a training-free approach to guide a pre-trained multimodal flow model to generate molecules with specified properties.  They do so by introducing  the guided velocity and guided rate matrix.

**Strengths:**

Training-free guidance makes it computationally advantageous over other flow-based molecular generation models.

**Weaknesses:**

1. The performance of this method may heavily depend on the quality and accuracy of the target predictor.
2. Conditional generation performance does not seem to be improved over the baselines

**Questions:**

1. Could you explain why the conditional independence of trajectory and target holds?
2. Could you elaborate on how the hyperparameter \alpha_t is chosen or tuned in practice?
3. Since the method depends on a predictor to guide property adherence, is there any evaluation of how predictor quality or accuracy impacts the overall performance of the guided generation?
4. There is typo in Table 4, for Vina score, TFG-Flow was falsely bolded.

---

### Meta-Review · Area_Chair_ZJgq · 2024-12-22

**Metareview:**

The paper explores guidance for the generation process of pre-trained unconditional generative models. It proposes TFG-Flow that focuses on cases where the data can contain discrete variables and when the generative model is of flow-matching trained on a mix of continuous and discrete variables, particularly MultiFlow. TFG-Flow deals with the bias in sampling of discrete guidance. They evaluate the model on molecular design that involves both continuous (pose) and discrete (atom types) variables.

The reviewers appreciated the relevance and broad applicability of the setup, the significance of the empirical results, and the extent of experiments.

On the other hand, there were questions regarding the assumptions for the type and motivation of the denoiser, run-time of the different models with respect to their size and type and the sampling being prohibitively expensive for real-world applications, the use of guidance from specifically-trained guidance models and empirical comparison to optimization-based drug-design methods.

The authors provided a thorough and continuous rebuttal and revision of the paper. This included experiments that rectified many of the concerns including those about the form of guidance and the baselines. In response to reviewer U4d6, they further majorly updated the paper on the necessary assumptions of their approach and importantly that of the base multiflow model which simplified the presentation and clarified some misconceptions and inconsistencies. While U4d6 appreciated the improvements they eventually stayed borderline and still leaned towards rejection.

The AC carefully considered the paper, the reviews, the rebuttal, and importantly the discussions during the rebuttal. The paper’s presentation went through a major refinement as a result of an active and admirable discussion between U4d6 and the authors. Such a major revision could potentially warrant another round of review but given the appraisal of other reviewers and the fact that the paper is pushing an active and exciting direction, the AC read the updated paper as a fresh reviewer and could not identify an error in the presentation. U4d6 was also eventually satisfied with the presentation. Therefore, the AC recommends acceptance as the paper has empirical merits.

**Additional Comments On Reviewer Discussion:**

The paper was reviewed by four expert reviewers covering all aspects of the paper. The reviewers’ take of the paper significantly improved during the rebuttal phase. There did not remain any major outstanding concern after the revisions. While three reviewers remained at borderline, one senior reviewer suggested clear acceptance.

---

### Decision · Program_Chairs · 2025-01-22

Accept (Poster)